# Grain growth of ice doped with soluble impurities

Qinyu Wang[1,2], Sheng Fan[3,4,5], and Chao Qi[1,2]

[1]Key Laboratory of Earth and Planetary Physics, Institute of Geology and Geophysics, Chinese Academy of Sciences, Beijing, 100029, China
[2]College of Earth and Planetary Sciences, University of Chinese Academy of Sciences, Beijing, 101408, China
[3]Department of Geology, University of Otago, Dunedin, New Zealand
[4]Department of Earth Sciences, University of Cambridge, Cambridge, UK
[5]RSC, Te Whanganui-a-Tara / Wellington, Aotearoa / New Zealand

**Correspondence:** Chao Qi (qichao@mail.iggcas.ac.cn)

**Abstract.** The grain size of polycrystalline ice affects key parameters related to the dynamics of ice masses, such as the rheological and dielectric properties of terrestrial ice flow as well as the ice shells of icy satellites. To investigate the effect of soluble impurities on the grain-growth kinetics of polycrystalline ice, we conducted annealing experiments on polycrystalline ice samples doped with different concentrations of KCl ($10^{-2}$, $10^{-3}$, $10^{-4}$ and $10^{-5}$ mol/L) or $MgSO_4$ ($10^{-2}$ and $10^{-5}$ mol/L). Samples were annealed for a maximum of 100 h at a hydrostatic confining pressure of 20 MPa (corresponding to a depth of about 2 km) and different constant temperatures of 268, 263, 258 and 253 K (corresponding to -5, -10, -15 and -20°C, respectively). After each experiment, images of a polished sample surface were obtained using an optical microscope equipped with a cold stage. With grain boundaries detected, grains were reconstructed from the images and an average grain size was determined for each sample. Normal grain growth occurred in all samples. Grain-size data are interpreted using the grain-growth model, $d^n - d_0^n = kt$ ($d$: grain size; $d_0$: starting grain size; $n$: grain-growth exponent; $k$: growth constant; $t$: duration). Values of the best-fit grain growth exponent, $n$, for all samples range from 2.6 to 6.2, with an average value of 4.7. Pure ice exhibits $3.1 \leqslant n \leqslant 4.6$, with an average value of 3.8. Above the eutectic point, soluble impurities enhance grain growth, as a melt phase is formed and it could provide a fast diffusion pathway. Below the eutectic point, soluble impurities impede grain growth probably via the formation of salt hydrates that could pin the grain boundaries. Close to the eutectic point, the grain growth of doped ice is similar to pure ice. Natural ice is impure, often containing air bubbles and soluble impurities, and is usually subjected to a hydrostatic pressure. Our data set will provide new insights to the evolution of grain size within and the dynamics of natural ice masses.

## 1 Introduction

Ice Ih, which forms glaciers and ice sheets on Earth, the polar ice caps on Mars, and the ice shells of icy moons (e.g., Europa), is of significant importance in planetary dynamics (e.g., Schubert et al., 1986; McKinnon, 1999; Rothery, 1999). Understanding

the rheological behavior of ice is crucial for interpreting the flow and dynamics of planetary ice masses (Bons et al., 2018; Barr and McKinnon, 2007; Ruiz, 2010). Previous deformation experiments on ice have revealed the influence of grain size on the mechanical behavior (e.g., Goldsby and Kohlstedt, 1997; Qi and Goldsby, 2021). In the grain-size sensitive deformation regime, ice samples with smaller grain sizes are mechanically weaker, resulting in reduced stresses under constant strain rates or increased strain rates under constant stresses (Goldsby and Kohlstedt, 2001). Grain size is not constant during deformation; instead, it represents a dynamic equilibrium between grain growth and grain-size reduction processes due to dynamic recrystallization. Previous laboratory experiments found that the rate of static grain growth driven by surface energy in ice was impeded by secondary phases, such as air bubbles and insoluble impurities, which often exist in natural ice (e.g., Nasello, 1992; Azuma et al., 2012; Kubo et al., 2009). In studies of glaciers, Alley et al. (1986a, b) found that soluble impurities had a more significant impact on the grain growth of ice, compared to bubbles and particles. However, some other studies reported that a high concentration of soluble impurities inferred from electrical conductivity measurements does not necessarily result in a slowdown of grain growth (Durand et al., 2006). Instead, pinning of grain boundaries by dust particle (insoluble impurities) accounts for the change in grain size (e.g., Weiss et al., 2002; Durand et al., 2006). Despite the wide presence of soluble impurities in natural ice, there is a scarcity of experimental research exploring the impact of these impurities on ice grain growth.

As an aqueous solution cools and freezes above the eutectic point (the temperature at which all the water in a solution solidifies into ice), soluble ions precipitate from the ice phase and accumulate at the grain boundaries in the form of concentrated saturated solutions. When the temperature drops below the eutectic point, the saturated solutions crystallize, producing different forms of solutes at different locations. For instance, sulfates may form hydrates at grain boundaries and within grains (Ohno et al., 2005, 2006; Sakurai et al., 2011). For chloride, although the majority of ions are expelled from the ice lattice to the grain boundaries, small quantities (approximately 1000 ppb) of ions can still be incorporated into the ice lattice (Montagnat et al., 2001). Previous experiments have demonstrated that soluble species, such as sodium ions ($Na^+$) and chloride ions ($Cl^-$), incorporated into the crystal lattice, can cause the formation of crystalline defects, thereby modifying physical properties of ice, including electrical conductivity (Petrenko and Whitworth, 1999). Varying concentrations of soluble impurities may exert distinct influences on the intricate processes of grain growth. De Achaval et al. (1987) conducted experiments on columnar ice samples made from highly concentrated NaCl solutions ($10^{-4}$ to $10^{-2}$ mol/L). Their results revealed a significant enhancement in the mobility of grain boundaries in the presence of these impurities at temperatures above the eutectic point of the NaCl solution. In contrast, Nasello et al. (2007) performed experiments on bicrystalline ice samples made from relatively low concentrations of KCl ($10^{-6}$ to $10^{-5}$ mol/L). They observed a decrease in grain boundary mobility with increasing concentrations of KCl.

In this contribution, we performed annealing experiments at different temperatures on polycrystalline ice with and without the presence of soluble impurities (potassium chloride, KCl, and magnesium sulfate, $MgSO_4$), and investigated the grain growth kinetics in these systems. We aim to provide new data and bring new insights to the complex interplay between soluble impurities and the process of ice grain growth.

## 2  Grain-growth model

Here, we focus on normal grain growth, driven by a decrease in grain-boundary energy. This mechanism of grain growth in geological materials has been investigated in previous studies, the details of which are well summarized in Evans et al. (2001), Bons et al. (2001) and Ohuchi and Nakamura (2007). In general, the evolution of grain size with time, $t$, follows a power-law relationship described by

$$d^n = d_0^n + kt, \tag{1}$$

where $d$ is the grain size after duration, $t$, $d_0$ is the starting grain size at $t = 0$, and $n$ is the grain-growth exponent. The grain-growth-rate constant, $k$, can be expressed as a function of temperature

$$k = k_0 \exp(-Q/\mathrm{R}T), \tag{2}$$

where $k_0$ is a constant, $Q$ is the activation energy for grain growth, R is the gas constant, and $T$ is the temperature. Theoretical analysis of grain growth in single-phase materials indicates that the exponent, $n$, is equal to 2 (Burke and Turnbull, 1952), with self-diffusion being the primary controlling factor. However, in the case of multi-phase materials, the value of $n$ is typically greater than 2 due to the presence of a second phase or impurities at the grain boundaries. Annealing experiments on synthetic polycrystalline ice have reported that the value of $n$ for bubble-free, pure ice was approximately 2 or 3 (Kubo et al., 2009; Azuma et al., 2012; Fan et al., 2023); the value of $n$ for ice doped with $1.7 \times 10^{-2}$ mol/L NaCl is about 4 (Jellinek and Gouda, 1969). However, in the case of polycrystalline ice with air bubbles, the value of $n$ ranged from 5.3 to 14.5, with an average of 8.3 (Azuma et al., 2012). In contrast, annealing experiments on natural ice samples cored from the Priestley Glacier, Antarctic, revealed $n \approx 50$, when $k$ was fixed to the value obtained from experiments on synthetic ice (Fan et al., 2023). In other minerals, such as olivine, the presence of impurities at grain boundaries, including partial melts, can slow down the grain growth and result in higher values of $n$ (Hiraga et al., 2010; Faul and Scott, 2006).

## 3  Methods

### 3.1  Sample preparation

Samples were prepared using a droplet solidification method (Wood and Walton, 1970; Goldsby and Kohlstedt, 1997; McCarthy et al., 2011). First, analytical standard solutions of KCl with a concentration of 0.01 mol/L or $MgSO_4$ with a concentration of 0.1 mol/L were mixed with ultra-pure water to prepare KCl solutions with concentrations of $10^{-5}$, $10^{-4}$, $10^{-3}$ and $10^{-2}$ mol/L or $MgSO_4$ solutions with concentrations of $10^{-5}$ and $10^{-2}$ mol/L, respectively. Second, the prepared KCl (or $MgSO_4$) solution was added to the container of a medical ultrasonic nebulizer, which generated a fine mist of droplets with a diameter less than 50 $\mu$m. An extension tube, with one end connecting to the outlet of the nebulizer and the other hanging into a liquid nitrogen dewar, transferred the mist to the liquid nitrogen at 77 K. The end of the tube was positioned a few centimeters above the liquid nitrogen level, so that droplets in the mist fell into liquid nitrogen and formed amorphous ice particles, without freezing on the

tube. After sufficient time (usually an hour), the slurry of liquid nitrogen and ice particles was sieved to collect ice particles $\leqslant$ 60 $\mu$m in diameter. Ice particles produced in this manner have salt distributed on the surface of the particles as fine filaments (Blackford, 2007). After the liquid nitrogen evaporated in the freezer, ice particles were cold-pressed in a stainless-steel mold at ~30 MPa and 243 K, to form a disc with a diameter of 25 mm and a height of roughly 6 mm. The disc was then vacuumed and sealed in a plastic bag using a food vacuum sealer. Next, the sealed bag was placed in a cold isostatic press (CIP) filled with antifreeze, and hydrostatically pressed at 100 MPa and 243 K for 15 min. During this high-pressure pressing, a phase transition from amorphous ice to ice Ih occurred, and ice particles were pressed into a fully densified ice sample. After fabrication, ice samples were stored in a storage dewar filled with liquid nitrogen.

## 3.2 Annealing experiments

Ice samples with different concentrations of impurities were grouped by temperature and desired dwell time. They were vacuumed and sealed in a plastic bag and placed in the CIP for annealing experiments. Annealing was conducted at a hydrostatic pressure of 20 MPa, corresponding to the pressure at the base of a 2-km glacier, and temperatures of 268, 263, 258 and 253 K (corresponding to -5, -10, -15 and -20°C, respectively), to a maximum duration of 100 h. Furthermore, annealing experiments with durations of 3.2 hours, 10 hours and 32 hours were respectively conducted to capture the rapid changes in microstructure and grain size in the early stage of annealing. One more group of KCl-doped samples (with $10^{-3}$, $10^{-4}$ and $10^{-5}$ mol/L KCl) were annealed at 258 K for both 3.2 hours and 100 hours. The results from this group of samples were compared with those from the original set of samples, to assess the reproducibility (see Figure A6 for details). The temperature inside the vessel of the CIP during annealing cannot be accurately measured due to its hermetic nature, and thus was calibrated prior to the experiments. Temperature calibration was done by measuring the temperatures inside the cold chamber (where the CIP was placed) and inside the antifreeze in the non-pressurized CIP (see Appendix A for details). The temperature differences were recorded at all targeted experimental temperatures. The calibration experiments conducted at different temperatures produced stable and repeatable results, as illustrated in Figure A1. Then the temperatures during annealing experiments can be calculated from the temperature inside the cold chamber. After the prescribed duration, the CIP was depressurized at temperature and quickly moved to a freezer at 243 K. In the freezer, samples were taken out from the press and the plastic bag, and then placed into a long-term storage dewar filled with liquid nitrogen within a few minutes.

## 3.3 Analysis of microstructure

After annealing, the grain size of each sample was determined using optical microscopy. A surface of each sample was prepared by polishing with sandpapers to a grit of 1200 in a freezer at 243 K. Subsequently, the polished surface was exposed to the air in the freezer for 10 minutes to deepen grain boundary grooves via sublimation. Using a couple drops of 0°C water, the prepared sample was frozen onto the bottom of an aluminium box, which was pre-cooled in the freezer. Then the box was filled with anhydrous ethanol (at 243 K) to cover the polished sample surface. Subsequently, the aluminium box was transferred to a cold stage set at 213 K coupled with an Olympus BX63 optical microscope. The ethanol could prevent frosting on the sample surface, while allow a penetration of light during the observation.

To cover an area with sufficient grains ($> 450$), at least 30 images for samples with finer grain sizes and up to 136 images for samples with coarser grain sizes were collected from the polished surface. For each sample, the images were merged into a large mosaic image covering an area of at least $1.5 \times 3.0$ mm. After corrected for shading and exposure, the mosaic image was processed using the 'cyto2' model in a computational framework called Cellpose (Stringer et al., 2021; Pachitariu and Stringer, 2022), to identify grain boundaries, as illustrated in Figure A2. It is important to note that manual correction may be necessary for some grain boundaries (more details in Appendix B ). With grain boundaries identified, the grain size for each sample was calculated using the image processing software ImageJ (Fiji) (Hartig, 2013). For each grain in the polished surface, the equivalent diameter of a circle with the same area of the grain was calculated as the size of the grain. Note that grain size determined this way represents the size of a 2-D cross section of a 3-D grain. For easy comparison with previous studies (e.g., Azuma et al., 2012), no scaling factor was applied to the equivalent diameter. The average grain size, $d$, and its standard deviation, $\sigma_d$, were then determined as the arithmetic mean and its standard deviation of the equivalent diameters, respectively. The normalized standard deviation ($\sigma_d/d$) can be utilized to evaluate the self-similarity of the grain-size distribution throughout the annealing (Faul and Scott, 2006). Grains smaller than 7.5 $\mu$m in diameter, which are noises from the grain-boundary identification, or located on the edge of the image were excluded in the calculation of the average grain size.

### 3.4 Analysis of grain-growth model

The value of $n$ in Eq. 1 was obtained from a non-linear least-squares fit of the grain size as a function of annealing time. To constrain the quality of this fit, we utilize a Monte-Carlo type approach. For a given set of grain sizes at a given temperature and for a given impurity concentration, random noise was added to the grain-size measurements. This noise was drawn from a log-normal distribution with a normalized standard deviation of 0.02, which roughly approximates the error in the grain-size measurement. These noisy data were then fit to Eq. 1 with a non-linear least-squares fitting method to $n$, $k$, and $d_0$. This process was repeated 500 times for each data set to generate a distribution of best-fit parameters. The final values of $n$, $k$, and $d_0$ were averaged from the 500 fitted values. In subsequent analyses, we also sought to compare values of $k$ among different impurity concentrations. However, this comparison is only meaningful if the value of $n$ is the same (more details in Section 5.1). Therefore, we conducted the entire fitting procedure again with fixed $n$ from 2 to 6, corresponding to the distribution of results of $n$, and calculated the goodness of these fits represented by the coefficient of determination, $R^2$. The results of this analysis are presented in Tables 1 and A1.

## 4 Results

### 4.1 Starting materials

Microstructures of the starting materials are illustrated as the 0-h samples in Figures 2 to 5. For a given composition and concentration, annealing experiments at different temperatures used the same batch of starting materials, such that the microstructures of 0-h samples presented in Figures 2 to 5 are the same. The microstructures of all starting materials display

consistent characteristics. Samples are fully-densified ice, with no apparent air bubbles or pores observed. Grains are generally equiaxed and polygonal shaped. The majority of grain boundaries are straight, while many are also slightly curved. Grain size generally exhibits a log-normal distributions. Pure ice and doped ice have a similar average grain size of approximately 50 $\mu$m (see Table 1).

## 4.2 Pure ice

After annealing under a confining pressure of 20 MPa, grains grow larger in all samples, with no air bubbles or pores observed. At the highest temperature (268 K), the microstructure is characterized by highly curved grain boundaries and very coarse grains ($\sim$1000 $\mu$m) surrounded by finer grains ($<$ 300 $\mu$m), after 100-h annealing (Figure 2(e)). The grain-size distribution remains log-normal, and in the sample annealed for 100 hours, there is higher frequencies at larger grain sizes (500 to 1000 $\mu$m). This variation in distribution is also revealed by the values of $\sigma_d/d$, which increases from 0.43 to 0.86 after 100-h annealing at 268 K. In contrast, at lower temperatures (263–253 K), the microstructure is dominated by straight grain boundaries and equiaxed polygonal grains (Figure 3(e), 4(e) and 5(e)). The grain-size distribution remains log-normal after annealing. The evolution of grain size with increasing annealing time for pure ice at all four temperatures is plotted in Figure 9. Grain growth is faster in samples annealed at warmer temperatures. Values of the grain growth exponent $n$ at different temperatures, determined from least-squares with the results summarized in Table 1, are similar among different temperatures, varying in the range of 3.1 to 4.6. As demonstrated in Table A1, the goodness of fits for pure ice is nearly identical when $n$ is fixed to either 3 or 4.

## 4.3 KCl-doped ice

Here we describe the microstructures, grain-size distributions (Figures 2 to 5), average grain sizes (Table 1) and grain-growth curves (Figure 6) of annealed samples with respect to temperature. The reproducibility was checked by repeated experiments on sample of KCl-doped ice annealed at 258 K (Figure A6). After annealing at 268 K, a temperature well above the eutectic point of $H_2O$ and KCl, 262.5 K (Brady, 2009), microstructures of samples with four different KCl concentrations are all characterized by roughly equiaxed grains, as illustrated in Figure 2(a) & (b). The grain-size distributions of most samples remain log-normal after annealing; while a few samples (such as the sample with $10^{-5}$ mol/L KCl annealed for 10 to 100 hours, as illustrated in Figure 2(b)) exhibit higher frequencies at grain sizes larger than the peak. For each sample, the mean grain sizes is very similar to the peak grain size of the distribution. The normalized standard deviations of grain sizes, $\sigma_d/d$, in samples with $10^{-2}$ and $10^{-3}$ mol/L KCl increase gradually with time from 0 to 32 hours, but decrease with time from 32 to 100 hours. Values of $\sigma_d/d$ in samples with $10^{-4}$ mol/L KCl increase gradually with time from 0 to 100 hours. Values of $\sigma_d/d$ in samples with $10^{-5}$ mol/L KCl increase with time from 0 to 10 hours, but decrease with time from 10 to 100 hours. $\sigma_d/d$ is a proxy for the homogeneity of grain sizes. At any given time during the annealing, grain sizes in all KCl-doped ice are coarser than those in pure ice, as illustrated in Figure 6(a). Grain sizes in samples with $10^{-5}$ mol/L KCl are largest, while samples with the other three concentrations have similar grain sizes.

After annealing at 263 K, a temperature similar to the eutectic point of 262.5 K, microstructures of samples with four different KCl concentrations are also characterized by roughly equiaxed grains, as illustrated in Figure 3(a) & (b). In samples

annealed for 100 hours, grain sizes vary greatly, showing many coarse grains (often $>600$ $\mu$m) surrounded by finer grains ($<200$ $\mu$m). The grain-size distribution remains log-normal after annealing. In the sample with $10^{-5}$ mol/L KCl annealed to 100 hours, the peak frequency is slightly lower and the tail towards larger grain sizes is longer compared to other samples.

The trends of the evolution of $\sigma_d/d$ for different concentrations in Table 1 correlate with grain-size distributions. For the two higher-concentration samples ($10^{-2}$ and $10^{-3}$ mol/L), values of $\sigma_d/d$ increase from $<0.5$ and stabilize at $\sim 0.6$ after a 10-hour annealing. In contrast, for the two lower-concentration samples ($10^{-4}$ and $10^{-5}$ mol/L), values of $\sigma_d/d$ stay $<0.5$ after 32-hour annealing and increase to $>0.7$ after 100-hour annealing. At any given time during the 100-h annealing, grain sizes of ice samples doped with different concentrations of KCl are similar to each other and also similar to the grainsize in pure ice, as illustrated in Figure 6(b). All values of $n$ are larger than that of the pure ice.

After annealing at 258 K, a temperature slightly below the eutectic point of 262.5 K, microstructures of samples with four different KCl concentrations are characterized by polygonal shaped grains, as illustrated in Figure 4(a) & (b). The grain-size distributions remain log-normal. The grain size distributions among the samples are similar to each other, suggested by the values of $\sigma_d/d$, keeping in the range between 0.4 and 0.6 for all samples. At any given time, grain sizes in all KCl-doped ice are slightly finer than those in pure ice, as illustrated in Figure 6(c). Although the sample with the highest concentration ($10^{-2}$ mol/L) has the smallest grain size, the differences in the grain sizes of doped ice are insignificant. All values of $n$ are larger than that of the pure ice.

After annealing at 253 K, a temperature well below the eutectic point of 262.5 K, microstructures of samples with four different KCl concentrations annealed to 10 hours are characterized by polygonal shaped grains and straight grain boundaries, but microstructures of samples annealed to 100 hours have curved grain boundaries, as illustrated in Figure 5(a) & (b). The grain-size distributions remain log-normal. The growth of grains are similar to each other, as the values of $\sigma_d/d$ vary within the range between 0.4 and 0.6 for all samples. Specially, in samples with higher concentrations ($10^{-2}$ and $10^{-3}$ mol/L), $\sigma_d/d$ exhibited minimal variation, with values ranging from 0.44 to 0.48 and 0.38 to 0.44, respectively. At any given time, grain sizes in ice doped with higher concentrations ($10^{-2}$ and $10^{-3}$ mol/L) are significantly finer than those in ice doped with lower concentrations ($10^{-4}$ and $10^{-5}$ mol/L), as illustrated in Figure 6(c). Moreover, the grain sizes of ice doped with lower concentrations are similar to pure ice.

### 4.4 MgSO$_4$-doped ice

Here we describe the microstructures, grain-size distributions (Figures 2 to 5), average grain sizes (Table 1) and grain-growth curves (Figure 7) of annealed samples with respect to temperatures. After annealing at 268 K, a temperature similar to the eutectic point of H$_2$O and MgSO$_4$, 269.5 K, microstructures of samples with two different MgSO$_4$ concentrations are all characterized by roughly equiaxed grains, as illustrated in Figure 2(c) & (d). The grain-size distributions remain approximately log-normal after annealing. Values of $\sigma_d/d$ in samples with both concentrations of MgSO$_4$ slightly increase with time.

After annealing at 263, 258 and 253 K, temperatures well below the eutectic point of 269.5 K, the behaviors of MgSO$_4$-doped samples are similar, and described together in this paragraph. Microstructures of samples with both MgSO$_4$ concentrations annealed to 10 hours are characterized by polygonal shaped grains and straight grain boundaries, but microstructures of samples

annealed to 100 hours have slightly curved grain boundaries, as illustrated in Figures 3(c) & (d), 4(c) & (d) and 5(c) & (d). The grain-size distributions remain approximately log-normal after annealing. The growth of grains is similar to each other, as the values of $\sigma_d/d$ vary within the range between 0.4 and 0.55 for all samples. At any given time during the annealing, grain sizes in MgSO$_4$-doped ice are always finer than those in pure ice, as illustrated in Figure 7(b), (c) & (d). At 263 K and 253 K, samples with high impurity concentration ($10^{-2}$ mol/L) exhibited slightly smaller grain sizes than the low-concentration sample ($10^{-5}$ mol/L). Overall, the diversity in grain size between different concentrations is not evident.

## 5 Discussion

### 5.1 Grain growth mechanism

Our experimental results yielded values of $n$ in the range of 2.6 to 6.2, with an average of 4.7, except for the sample with $10^{-5}$ mol/L MgSO$_4$ annealed at 258 K. Fitting of data from this sample suggests $n = 1.1$, which is lower than the theoretical value of 2. We think this low value of $n$ may be attributed to the slow growth in the first 10 h. Because experiments are not perfect, such variations in the fitted values of $n$ were also reported in previous studies (Azuma et al., 2012). The variation of $n$ with temperature and impurity concentration is illustrated in Figure 8. $n$ values of doped ice are relatively greater than those of pure ice. At a given impurity concentration, no systematic change of $n$ with temperature is found. At 258 and 253 K, $n$ tends to decrease with increasing impurity concentration in KCl-doped ice; while at 268 and 263 K, no trend of $n$ with concentration is found. In the samples that a trend in $n$ was observed, the variations in $n$ are in the same magnitude as the uncertainties in $n$ due to sample-to-sample variation, $>1$ variation in $n$ (Azuma et al., 2012). While no systematic changes were observed in the value of $n$ as the solute concentration or temperature changed, we cannot rule out that the uncertainty in the value of $n$ may be higher than the systematic changes due to solute concentration and temperature. Thus, at this stage, the average value of $n$ can be regarded as representative of all the samples.

Theoretical analysis of grain growth in a single-phase material yields $n = 2$ (Burke and Turnbull, 1952). However, experimental observations often reveal values of $n$ exceeding 2. This phenomenon can be attributed to the influence of impurities and/or pores (air bubbles) along grain boundaries, that are difficult to eliminate during the sample preparation. When pores exist, the migration velocity of grain boundaries is significantly influenced by the rate at which pores can be dragged, a process that depends on the mass-transport rate through or around the pores (Shewmon, 1964). When a secondary phase exists, theoretical predictions indicate that value of $n$ may be 2 or 3 for transport through the secondary phase, 3 or 4 for transport by diffusion through the crystal lattice or through phase boundaries of the primary phase, respectively (Brook, 1976; Evans et al., 2001). For the case of impurities, in metals and ceramics, impurity segregation into grain boundaries can lead to larger values of $n$ (Hiraga et al., 2010). It is worth noting that experimental results often indicate larger values of $n$ compared to theoretical predictions. For ice samples containing pores, values of $n$ have been found to range from 6.2 to 11.1 (Azuma et al., 2012). In a polymineralic system of forsterite + enstatite, values of $n$ were reported to be approximately 5 (Hiraga et al., 2010). Similarly, in a dual-phase steel, values of $n$ were reported to be approximately 5 (Najafkhani et al., 2019).

Based on the knowledge from literature, the average value of $n$ in this study suggests that the grain growth occurred in a multi-phase system. The secondary phase is melt, that is, saturated salt solution, for annealing at temperatures above the eutectic point, and salt hydrate for annealing at temperatures below the eutectic point. The most possible theoretical explanation for the control of our grain growth process is mass transport via surface diffusion, corresponding to $n = 4$. For further analyses and comparisons, fitting with a fixed value of $n$ is necessary. Therefore, fitting with $n = 4$ was applied to all the data, enabling us to calculate the growth constant $k$. The results of this analysis are presented in Table 1. The goodness of fits ($R^2$ in Table 1) spans a range from 0.78 to 0.99, with the majority exceeding 0.95, underscoring the suitability of $n$ equaling 4 for fitting our data effectively.

## 5.2 Grain growth rate

Figures 6 and 7 show the grain-size evolution with annealing time for all samples at four different temperatures. In most samples, the grain size increases accompanied by a deceleration in growth rate over time. Notable grain size coarsening occurred within the initial 32 h, with the most substantial growth observed in the initial 3.2 h. Subsequently, from 32 hours to 100 hours, the grain growth is relatively slow. Some features of the microscopic processes of grain growth in pure ice and doped ice in our experiments can be explored based on a simple growth model. In this model, the velocity of grain-boundary migration, $v_b := g\frac{\partial d}{\partial t}$, is controlled by diffusion of the rate-limiting species along grain boundaries, which is expressed as (Hillert, 1965)

$$v_b = MP, \tag{3}$$

where $g$ is a geometric factor, $M$ is the grain-boundary mobility, and $P$ is the driving force. The grain-boundary mobility, which is limited by self-diffusion of the rate-limiting species in the grain boundaries, is a function of composition and temperature.

The driving force for normal grain growth is expressed as (Atkinson, 1988; Evans et al., 2001)

$$P = \gamma_b \left( \frac{1}{r_1} + \frac{1}{r_2} \right), \tag{4}$$

where $\gamma_b$ denotes the grain boundary energy, and $r_1$ and $r_2$ are the principal radii of the grain-boundary curvature. The difficulty in utilizing this equation to understand growth kinetics in real materials lies in relating the local curvature to grain size. Hillert (1965) argued that grains smaller than a critical grain size, $d_c \approx \frac{9}{8}d$, should have convex boundaries and will therefore shrink while others grow.

On the other hand, while the grain growth rate is primarily governed by the diffusion of ionic species, it is also subject to external resistance factors, including impurity resistance and bubble pinning. Historically, it was believed that bubbles would have minimal impact on grain boundary migration in glaciers (Duval, 1985; Alley et al., 1986b). Nevertheless, based on in-situ observations and experimental findings, the gradual grain growth observed in polar ice cores remains significantly influenced by the speed of bubble migration (Azuma et al., 2012; Kipfstuhl et al., 2006, 2009). Following annealing, no bubbles were observed in the samples. However, we cannot exclude the possibility that the bubble size falls below our detection limit. It is inevitable that some bubbles would persist during the cold pressing due to the lack of high vacuum. The application of a 100-MPa pressure during subsequent hydrostatic pressing further compressed these bubbles; however, complete closure of the

bubbles might not be achieved. The migration rate of grain boundaries will be constrained by the maximum bubble migration rate, inversely proportional to bubble size (Azuma et al., 2012). Therefore, as a grain boundary migrates, it will be dragged by more bubbles (if they exist) and/or larger bubbles if bubbles coalesce, and the migration rate of grain boundaries will decrease.

### 5.3    Grain growth of pure ice

    Before looking into the effects of soluble impurities, it is prudent to compare our findings from pure ice samples with those
available in the literature. In our study, the evolution of grain size with increasing time for pure-ice samples annealed at all four temperatures is plotted in Figure 9(a). The values of $n$ determined by least-square fitting for these pure ice samples range from 3.1 to 4.6, as summarized in Table 1. Previous studies examining the normal grain growth of pure ice, employing various methods, have produced varying results. For instance, Jellinek and Gouda (1969) reported an $n$ value of approximately 3.3 by annealing two-dimensional ice with a thickness of 500 $\mu$m, which was frozen on glass plates, at temperatures of 237 to
270 K. Compared to polycrystalline ice, the growth of two-dimensional ice crystals is influenced by the surface diffusion of water molecules (Azuma et al., 2012). Fan et al. (2023) annealed coarse-grained ice ($\sim$900 $\mu$m) at 273 K and found $n$ values ranging from 2 to 3. This experiment performed annealing at a sole temperature, limiting its ability to assess the impact of temperature on grain growth. Azuma et al. (2012) conducted an extensive study on the normal grain growth of ice. By annealing polycrystalline ice prepared through the phase-transformation method within a temperature range of 243 to 268 K,
they obtained $n$ values ranging from 1.7 to 3.0, with an average close to 2. In our study, the values of $n$ are generally greater than both the theoretical value and the values obtained from the above experiments, particularly (Azuma et al., 2012). It is worth noting that many previous studies have directly fit $d^2$ (grain cross-section area) with time by assuming that pure ice follows the theoretical value of $n = 2$ (e.g., Gow, 1969; Azuma and Higashi, 1983; Paterson, 1994, pp. 38–40). To compare the growth rate of pure-ice samples between our study and the study of Azuma et al. (2012), we conducted fits with $n$ fixed
to 2 for the pure-ice data, as illustrated in Figure 9(b), allowing us to calculate the growth constant, $k$. Figure 9(c) presents a comparison of $k$ for bubble-free pure ice between Azuma et al. (2012) and our study. A fit of our four data points of pure ice to Eq. 2 yields an activation energy $Q \approx 46\pm25$ kJ/mol, which is very close to the activation energy obtained by Paterson (1994, pp. 39). Notably, in the study by Azuma et al. (2012), the grain growth rate (represented by $k$) was markedly faster, two to three orders of magnitude greater, than that observed in our study at a similar temperature range. Furthermore, the value of
$Q$ in their study is approximately $113 \pm 8$ kJ/mol. Given that Azuma et al. (2012) provided the most comprehensive data set to date on the normal grain growth of polycrystalline ice, it is imperative to gain insights into the factors contributing to the differences in $n$ and $k$ values observed in our study.

    Firstly, one notable difference between our study and that of Azuma et al. (2012) lies in the methodology employed for sample preparation. Azuma et al. (2012) utilized a phase-transformation method to produce fine-grained samples, while we
employed a droplet solidification method. In the phase-transformation method used by Azuma et al. (2012), the following procedure was followed: (1) Ultra-pure water was frozen in a container from bottom up to form polycrystalline ice. (2) The sample was cut and pressurized to 300 MPa in a Teflon chamber filled with silicone oil at 223 K for 2 h to complete the transformation from ice Ih to ice II. (3) The sample was then rapidly depressurized to 100 MPa, which was maintained for

1 h, to complete the transformation from ice II back to ice Ih. This method, initially introduced by Stern et al. (1997), relies on the volume change during phase transformation to produce strain-induced recrystallization, resulting in a very fine-grained microstructure. It is worth noting that this technique is typically limited to temperatures below 220 K (Stern et al., 1997). The higher temperature used in the study by Azuma et al. (2012) could partly account for the larger initial grain sizes in their experiments.

In the context of our sample preparation procedure, it is important to acknowledge the potential for air bubbles to be trapped within the samples. As described in the preceding section, our method involves sealing an ice disc in a plastic bag, and compression at a confining pressure of 100 MPa. This method is frequently used by the rock-deformation community, especially on olivine (e.g., Hansen et al., 2012). Recent developments in experimental rock physics have revealed that this fabrication method cannot drive air bubbles out of the sample but only compress them to sizes too small to be observed, and other methods, such as vacuum sintering and evacuated hot-pressing, are necessary to produce bubble-free samples (e.g., Koizumi et al., 2010; Meyers et al., 2017). Thus, for our samples, it is possible that the pores were not completely closed up, and air molecules might still have been confined within these pores. Although we did not observe any air bubbles in our samples either during preparation or annealing, it is important to consider the possibility that air bubbles, with sizes below the detection limit of our optical microscope, might have been trapped along grain boundaries in the samples. These bubbles, while not observable, could still exert a pinning effect when grain boundaries migrate. This pinning effect is the most plausible explanation for the larger values of $n$ observed in our samples. To test this hypothesis, a couple of samples were warmed to above 233 K in ethanol under the optical microscope. At such warm temperatures, ice became softer and dissolved more in ethanol. Some air bubbles appeared in our samples (see Figure A3).

Secondly, another difference between our study and that of Azuma et al. (2012) lies in the pressure conditions. While Azuma et al. (2012) conducted their annealing experiments at one atmosphere, we performed ours at a hydrostatic pressure of 20 MPa. In many materials, including stainless steel and magnesium oxide, studies have observed that both the grain growth rate and grain boundary diffusion coefficient decreased at higher pressures when compared to lower-pressure conditions (e.g., Farver et al., 1994; Krawczynska et al., 2018). However, ice behaves differently at high-pressure conditions compared to most oxides. Noguchi et al. (2016) found that ice's self-diffusion coefficient has a negative activation volume, implying that the self-diffusion rate increases with increasing pressure. In a previous annealing experiment conducted on 1 mm-thick slices of both natural and lab-made ice, Azuma and Higashi (1983) determined that the grain growth rate of ice was higher in samples annealed at high pressure (50 MPa) than in those annealed at atmospheric pressure. Furthermore, Breton et al. (2016) discovered, through deformation tests conducted at both 20 MPa and one atmosphere, that the migration rate of grain boundaries significantly increased at higher confining pressures. Although we think the observed increase with increasing confining pressure could be attributed to the closure of bubbles and the prevention of microcracks, it is recognized that the grain growth rate should probably increase with increasing confining pressure. Hence, the presence of confining pressure is unlikely to be the cause for the larger values of $n$ in our study.

Another significant difference between our study and that of Azuma et al. (2012) is the duration of the annealing experiments. Azuma et al. (2012) conducted their experiments for much longer durations compared to our study. In the study by Azuma

et al. (2012), bubble-free pure ice samples were annealed for a maximum duration of 840 h. In our study, due to the low
temperatures, the seals for the CIP vessel cannot last longer than 100 h. Consequently, our longest annealing experiments were
limited to 100 h. The limited duration of annealing in our study indeed resulted in a limited set of data points, which could
potentially have an effect on the calculated $n$ values.

At last, it is worth highlighting another distinction between our study and that of Azuma et al. (2012) concerning the chemical
environment of samples. In the study by Azuma et al. (2012), their samples were in contact with silicone oil during the phase
transformation at high pressures. In contrast, our samples were consistently sealed in plastic bags, which prevented any contact
with antifreeze or other liquids. The potential impact of silicone oil on ice at high pressures remains unclear and warrants
further investigation. The presence of such a substance during the phase transformation could introduce additional variables
that might affect the grain growth process and subsequently influence the calculated values of $n$. Therefore, the role of silicone
oil in these high-pressure conditions merits a closer examination.

## 5.4  Effect of soluble impurities

Based on the comparison of results for pure ice samples between our study and those in the literature, it is possible our samples
were affected by air bubbles that are too small to be detected. The presence of air bubbles could induce large uncertainties in the
fitting of $n$ (Azuma et al., 2012), which may overshadow the effect of different impurities contents on $n$, if existed. However, in
this study, we have to make inferences based on the available data set, while acknowledging the uncertainties. Given the same
experimental conditions, we assume the effect of air bubbles should apply equally to pure and doped ice samples, and thus, we
can investigate the effect of soluble impurities on grain growth by comparing samples with different impurity concentrations.
As elucidated in the previous subsections, the values of $n$ suggest that the grain growth occurred in a multi-phase system.
We have not observed any evident systematic trends in $n$ with changes in temperature or solute concentration. We tentatively
assume that the grain growth in all doped ice samples can be considered as controlled by the same mechanism. As illustrated in
Figure A7, the majority of fitted values of $n$ lie around 4 and 5. And, as illustrated in Table A1, $n = 4$ provides good fits for all
samples. We take $n = 4$ as a representative value for all samples in the following discussions. Applying $n = 4$, the theoretical
value for growth controlled by mass transport via surface diffusion, to all data, we get relatively good fits and values of $k$, as
illustrated in Figure 10. As described in Results section, relative to pure ice, doped ice samples exhibit different growth rates
at different temperatures. In the following discussions, doped ice samples were classified into three groups by their annealing
temperature: above, below and at eutectic temperature. For each group, the growth rates of doped ice were compared with that
of pure ice, so that the effect of soluble impurities can be revealed.

### 5.4.1  Above the eutectic temperature

This group of samples pertains to those annealed at temperatures at least 2 K above the eutectic point, specifically the KCl-
doped ice annealed at 268 K. Hence, our subsequent discussion is exclusively centered on KCl-doped ice. As illustrated in
Figure 6(a), grain sizes in doped ice samples are larger than those observed in pure ice. This trend is further emphasized in
Figure 10(a), where the values of $k$ for various concentrations of doped ice exceed those of pure ice. These observations align

with the findings of previous studies, such as Jellinek and Gouda (1969), who investigated two-dimensional ice doped with NaCl ($2.2 \times 10^{-2}$ mol/L), and De Achaval et al. (1987), who studied columnar ice doped with NaCl ($10^{-4}$ to $10^{-2}$ mol/L). When water freezes from salt solutions, most salt ions are rejected from the solid phase and become concentrated in the liquid phase (Conde et al., 2017). This process results in the formation of "melt pockets" of brine form along grain boundaries and at triple junctions in polycrystalline ice. While the presence of melt along grain boundaries could theoretically exert a dragging force on grain boundary migration, it is essential to recognize that the melt phase also acts as a rapid diffusion pathway. When the diffusion within the melt phase is significantly faster than the diffusion within the ice, the enhancement of diffusion predominates over the pinning effect on grain boundary migration. Consequently, the presence of a melt phase actually promotes grain growth rather than inhibiting it. A similar phenomenon was reported by Hammonds and Baker (2018), who observed significantly larger grain sizes in $H_2SO_4$-doped ice compared to pure ice after deformation at temperatures above the eutectic point. Therefore, at temperatures above the eutectic point, grain growth is accelerated by soluble impurities when compared to pure ice, due to melt-enhanced diffusion.

### 5.4.2 Below the eutectic temperature

The second group were samples annealed at temperatures at least 2 K below the eutectic point. This group includes the KCl-doped ice annealed at 258 K and lower temperatures and the MgSO$_4$-doped ice annealed at 263 K and lower temperatures. Ice doped with KCl and MgSO$_4$ exhibit slightly different behaviors, which will be examined separately. In the case of KCl-doped ice annealed at 258 K, grain sizes in doped ice samples are smaller than those observed in pure ice, as illustrated in Figure 6(c). However, except for the sample doped with $10^{-2}$ mol/L KCl, the grain sizes of doped ice fall roughly within the uncertainty range of those observed in pure ice. A similar observation is evident in Figure 10(a), where the values of $k$ for the three lower concentrations ($10^{-5}$ to $10^{-3}$ mol/L) of doped ice are slightly lower than those of pure ice but remain within the uncertainty range. For KCl-doped ice annealed at 253 K, samples with lower concentrations ($10^{-5}$ and $10^{-4}$ mol/L) behave similarly to pure ice, while samples with higher concentrations ($10^{-3}$ and $10^{-2}$ mol/L) exhibit significantly smaller grain sizes than pure ice, as illustrated in Figure 6(d). This same trend is observed in Figure 10(a), where the values of $k$ for the two higher concentrations of doped ice are notably lower than those of other samples including pure ice, which are similar between each other. For MgSO$_4$-doped ice annealed at all three temperatures, grain sizes in doped ice samples are consistently smaller than those observed in pure ice, as illustrated in Figure 7. This trend is further emphasized in Figure 10(b), where the values of $k$ of doped ice are lower than those of pure ice. At 263 and 253 K, samples with the higher concentration ($10^{-2}$ mol/L) exhibit smaller grain sizes and lower $k$ values than samples with the lower concentration ($10^{-5}$ mol/L), while at 258 K, the trend is reversed. However, ice doped with different concentrations are very similar at all temperatures, with grain sizes and $k$ values fall within the uncertainty ranges of each other.

As previously mentioned, the solubility of salt in ice is exceedingly low. When the temperature drops below the eutectic point, the melt phase solidifies. In the case of impurities like salts, the brine freezes into salt hydrates. These hydrates take the role of a secondary phase, pinning the grain boundaries when they migrate, and thereby further impeding the process of grain growth. Despite ice's pronounced repelling effect on salt ions, prior experiments have demonstrated that chlorides, such as KCl

and NaCl, can be uniformly distributed as solutes within ice lattice, at exceedingly low concentrations ($10^{-5}$ to $10^{-4}$ mol/L) (Montagnat et al., 2001; Yashima et al., 2021). However, sulfates have been found to concentrate along grain boundaries rather than within the ice lattice (Montagnat et al., 2001). This difference in the solubility of ions in ice lattice results in the different behaviors of KCl and $MgSO_4$ in our study. In ice doped with $MgSO_4$, salt hydrates form and impede grain growth, regardless of the impurity concentration being low ($10^{-5}$ mol/L) or high ($10^{-2}$ mol/L). In contrast, in ice doped with KCl, salt hydrates only form and impede grain growth when the impurity concentration exceeds $10^{-3}$ mol/L.

### 5.4.3 At the eutectic temperature

Samples annealed within 2 K above and below the eutectic point were categorized as the group annealed at the eutectic temperature. This group comprises KCl-doped ice annealed at 263 K and $MgSO_4$-doped ice annealed at 268 K. The behavior of samples in this group, relative to pure ice, falls in between the behaviors of the previous two groups. For both KCl and $MgSO_4$, the grain sizes in doped ice are indistinguishable compared to those of pure ice, as illustrated in Figures 6(b) and 7(a). This similarity is further confirmed in Figure 10, where the values of $k$ of doped ice closely resemble those of pure ice. Although the annealing temperatures for both compositions deviated slightly from the eutectic points (0.5 K above the eutectic point for ice + KCl system and 1.5 K below the eutectic point for ice + $MgSO_4$ system), uncertainties in temperature measurements and fluctuations during experiments could possibly lead to the samples experiencing temperatures both above and below the eutectic point. This nuanced temperature exposure results in a behavior that falls in between the two cases discussed previously. In essence, there appears to be neither enough melt to significantly enhance grain growth nor enough salt hydrates to impede grain growth. As a result, the doped ice samples exhibit grain growth rates similar to those of pure ice.

### 5.4.4 A factor for the effect of soluble impurities

As discussed in the previous paragraphs, the influence of soluble impurities on grain growth varies with temperature relative to the eutectic point: (1) At temperatures above the eutectic point, soluble impurities enhance grain growth by forming a melt phase that provides a rapid diffusion pathway. (2) At temperatures below the eutectic point, soluble impurities, when present above the threshold concentration that can be accommodated in the ice lattice, impede grain growth by forming salt hydrates that slow down grain boundary migration. (3) At temperatures close to the eutectic point, soluble impurities make no significant impact to grain growth compared to pure ice. To facilitate the practical application of our findings to natural ice masses, we can calculate the apparent activation energy for the grain growth of ice with soluble impurities. A fit of data of ice doped with $10^{-2}$ and $10^{-3}$ mol/L KCl, sufficient concentrations to have an impact on grain growth at temperatures below the eutectic point, to Eq. 2 yields $Q_{KCl} = 161 \pm 25$ kJ/mol. A fit of data of ice doped with $10^{-2}$ and $10^{-5}$ mol/L $MgSO_4$ to Eq. 2 yields $Q_{MgSO_4} = 142 \pm 57$ kJ/mol. Lines for the two fits are plotted in Figure 10. Importantly, both values of the "apparent activation energy" for these impure ice fall within the uncertainty ranges of each other, suggesting that the effects of both KCl and $MgSO_4$ can be considered together when assessing their impact on grain growth in ice.

We propose that the grain-growth constant of ice with soluble impurities, $k_{\mathrm{im}}$, can be related to that of pure ice, $k_{\mathrm{pure}}$, in this form

$$k_{\mathrm{im}} = F(T, Te)k_{\mathrm{pure}}, \tag{5}$$

where $F$ is a function of temperature, $T$, and eutectic point, $Te$, representing the effect of impurity. Since $k_{\mathrm{im}} \approx k_{\mathrm{pure}}$ at the eutectic point, $F(T = Te, Te)$ must equal to 1. Thus, we can construct factor $F$ in this form

$$F(T, Te) = \exp\left(A_0\left(\frac{T}{Te} - 1\right)\right), \tag{6}$$

where $A_0$ is a factor to be determined by fitting. A fit of the pure-ice data to Eq. 2 yields $Q_{\mathrm{pure}} = 68 \pm 47$ kJ/mol and $k_{0\mathrm{pure}} = 6.19 \times 10^{-8}$ m$^4$/s. Taking the fitting results of $k_{\mathrm{pure}}$ into Eq. 5, we can perform a global fit of data of ice doped with $10^{-2}$ and $10^{-3}$ mol/L KCl and all data of ice doped with MgSO$_4$. A linear least-squares fit yields $A_0 = 39 \pm 11$. Consequently, we obtain the relationship:

$$k_{\mathrm{im}} = \exp\left(39\left(\frac{T}{Te} - 1\right)\right)k_{\mathrm{pure}} = \exp\left(39\left(\frac{T}{Te} - 1\right)\right)k_{0\mathrm{pure}}\exp\left(-\frac{Q_{\mathrm{pure}}}{RT}\right). \tag{7}$$

By substituting $Te = 262.5$ K for ice + KCl and $Te = 269.5$ K for ice + MgSO$_4$ into Eq. 7, we can determine the $k$ values for impure ice, as plotted in Figure 11. Results of Eq. 7 align well with the data points.

## 5.5   Implications for natural ice

Although Azuma et al. (2012) provided the most comprehensive data set to date on the normal grain growth of polycrystalline ice, applying their grain growth law to natural ice has not proven entirely successful. For instance, Behn et al. (2021) used the grain growth parameters from Azuma et al. (2012) in their study on modelling the grain-size evolution of ice. They found that their model, using $n = 2$ of the grain growth of bubble-free pure ice, overpredicts the grain size in the GISP2 ice core. On the other hand, the average value of $n$ of the bubble-rich ice in Azuma et al. (2012), which was 8.3, was too high for their model. Behn et al. (2021) suggested that a value of $n$ between 3 and 4 aligns well with the flow law parameters for grain-boundary sliding of ice. As discussed in earlier subsections, one potential limitation of the results from Azuma et al. (2012) lies in the way their samples were prepared. Their bubble-free ice samples were fabricated by a phase-transformation method, and their bubble-rich ice samples were not formed at a hydrostatic pressure. However, ice in glaciers and ice sheets is formed through processes involving compacting and hydrostatic compression. This process is very similar to our sample preparation process. We propose our grain growth law for ice with soluble impurities could fit the grain-size evolution of natural ice well. Taking either $n = 4$ from the theory or $n = 4.7$ from the average of our experimental data into the model of Behn et al. (2021), stress exponents of 2.9 and 2.7 for ice flow are obtained. These values are reasonable for ice that undergoes deformation through a combination of dislocation creep and grain-boundary sliding.

Based on the experimental findings, soluble impurities exhibit a discernible influence on grain growth. Nonetheless, given the substantial disparities in temperature, impurity types, and concentrations between our experimental conditions and those of natural glaciers, a more meticulous discussion is warranted when extrapolating our results to natural glacier settings. Cl$^-$

and $SO_4^{2-}$ constitute two of the most prevalent anions in polar glaciers, while $Na^+$, $Mg^{2+}$, $Ca^{2+}$, and $K^+$ are the predominant alkali metal cations in Earth's glaciers (Obbard and Baker, 2007). In this investigation, we have chosen KCl and $MgSO_4$

as representatives of chloride salts (hereinafter specifically referred to as chloride salts of the above four ions) and sulfates (hereinafter specifically referred to as sulfates of the above four ions), respectively. The impacts of KCl and $MgSO_4$ on ice grain growth can be attributed to their eutectic points and solubility within the crystal lattice of ice. In extrapolating our results to the influence of chloride or sulfate on ice grain growth, it is imperative to consider these two aspects. We suggest that it can be evaluated in two scenarios: above the eutectic point and below the eutectic point.

(1) In scenarios above the eutectic point, our dataset is relatively limited, because this study only includes data related to KCl-doped ice annealed at 268 K (-5°C). Our results reveal a significant promotion of grain growth by soluble impurities at temperatures above the eutectic point. This phenomenon is also observed in NaCl-doped ice (De Achaval et al., 1987). Except for KCl, other chloride salts exhibit relatively low eutectic points (approximately 223 K to 252 K, according to Baumgartner (2009)), potentially nearing or even falling below the temperature of some glaciers (Cuffey et al., 2000). In other words, most

glaciers have temperatures above the eutectic point of chloride salts. Nevertheless, the lowest concentration utilized in our experiments ($1\times10^{-5}$ mol/L) remains higher than the total impurity concentration in the ice sheet, estimated at about $1\times10^{-7}$ to $1\times10^{-5}$ mol/L (Cuffey and Paterson, 2010, pp. 33–34). Applying our experimental results directly to glaciers with low impurity concentrations poses challenges. Consequently, extra caution should be paid when extrapolating our results to lower impurity concentrations in glaciers.

(2) At temperatures below the eutectic point, we hypothesize that salt hydrates will precipitate at grain boundaries, impeding grain growth only when the concentration surpasses the solubility in the crystal lattice of ice. Given that $Cl^-$ ions can integrate into the interior of the ice crystal lattice, chloride salts exhibit slightly greater solubility in ice than sulfate salts. Specifically, the solubility of KCl is approximately 2 to $3\times10^{-5}$ mol/L (Yashima et al., 2021), while NaCl's solubility ranges from $3\times10^{-5}$ to $1\times10^{-4}$ mol/L (Montagnat et al., 2001; Gross et al., 1977). For $Ca^{2+}$ and $Mg^{2+}$ with smaller ionic radii, we assume

that the solubility of their corresponding chlorine salts ($CaCl_2$ and $MgCl_2$) in the ice lattice is of the same order as KCl and NaCl, that is, at least greater than $1\times10^{-5}$ mol/L. Thus, the solubility of common chloride salts may generally exceed the average content range of glaciers ($1\times10^{-7}$ to $1\times10^{-5}$ mol/L). Therefore, the effect of chloride salts on grain growth in colder glaciers is minimal. In contrast to KCl, grain growth remains impeded in the ice sample doped with $MgSO_4$ even at the lowest concentration ($1\times10^{-5}$ mol/L). We assume the solubilities of sulfates are similar to the solubility of sulfate ion, which may

be lower than $2\times10^{-6}$ mol/L (Iliescu and Baker, 2008). Additionally, the eutectic points of sulfates are generally high ($>$ 268 K), placing glaciers below these temperatures. In ice deposited during Pleistocene glacial climates with higher impurity concentrations ($10^{-6}$ to $10^{-5}$ mol/L, according to Cuffey and Paterson (2010, pp. 33–34)), sulfates may hinder grain growth. This conclusion could establish a connection between reduced grain size in ice cores and elevated concentrations of soluble impurities (sulfate and chloride ions) (e.g., Alley et al., 1986a; Langway et al., 1988; Paterson, 1991).

## 6 Conclusions

- Grain growth occurred in pure ice and polycrystalline ice doped with $10^{-5}$ to $10^{-2}$ mol/L KCl or $MgSO_4$ in annealing experiments conducted at temperatures between 253 K and 268 K and a confining pressure of 20 MPa. The values of the best-fit grain growth exponent $n$ lie between 2.6 to 6.2.

- The grain growth exponent for pure ice samples ranged from 3.1 to 4.6. Compared to previous studies, we observed that the grain growth rate in the laboratory is lower than that of artificial pure ice; and the activation energy for grain growth is close to that of glacial ice and artificial ice containing smaller bubbles. This may be because some bubbles still exit in our samples, although they are compressed by the confining pressure to sizes below the detection limit of our optical microscope.

- Above the eutectic point, grains doped with soluble impurities exhibit a higher grain growth constant, compared to pure water ice, i.e., faster grain growth rate. Below the eutectic point, ice doped with a specific concentration of soluble impurities manifests a smaller grain growth constant, compared to pure water ice, i.e., slower grain growth rate.

- The enhancement in grain growth at temperatures above the eutectic point could be attributed to the formation of a molten phase by the doped salt. The inhibition in grain growth at temperatures below the eutectic point may be attributed to the formation of hydrates at grain boundaries.

- When the concentration of soluble impurities exceeds the limit that can be accommodated in the ice lattice, the effect of soluble can be described by a factor, $F$, in the following equation:

$$k_{\mathrm{im}} = F k_{\mathrm{pure}} = \exp\left(A_0\left(\frac{T}{Te} - 1\right)\right) k_{\mathrm{pure}}, \tag{8}$$

where $A_0$ is determined to be $39 \pm 11$ based on our experimental data.

- For future research, it is essential to develop a sample preparation method capable of producing fine-grained ice without introducing air bubbles. Moreover, although the two salts used in our experiments exhibit similar growth kinetics, different soluble impurities could have different effects and will be investigated in future studies.

*Data availability.* Data are available on an online data repository: Data of "Grain growth of ice doped with soluble impurities". figshare. Dataset. https://doi.org/10.6084/m9.figshare.24212103.v2

*Author contributions.* CQ designed the research. QW performed experiments and analyses. All authors participated in the interpretation of results. QW wrote the first draft, and all authors edited the manuscript.

*Competing interests.* The authors declare that they have no conflict of interest.

*Disclaimer.* TEXT

*Acknowledgements.* The authors thank to Takehiko Hiraga and his group for stimulating discussions. Helpful comments from Felicity Mc-Cormack, Lisa Craw and Christopher Gerbi are really appreciated. This work was supported by a NSFC grant 41972232 (to CQ) and the Key Research Program of the Institute of Geology and Geophysics, CAS, grant IGGCAS-201905 (to CQ).

## Appendix A: Temperature calibration

During an annealing experiment, while the cold isostatic press (CIP) is sealed and pressurized, a direct measurement of the temperature inside the vessel of the CIP is unfeasible with our experiment setup. To overcome this limitation, a temperature calibration procedure was conducted. This calibration involved monitoring the temperature of the antifreeze inside the unsealed CIP vessel, as illustrated in Figure A1. This calibration process mirrored the steps of a typical annealing experiment. The antifreeze was pre-cooled for a minimum of 2 hours within the CIP vessel situated in the low-temperature test chamber. The pre-cooling temperature was set 1 to 2 K below the designated annealing temperature. Once the antifreeze temperature reached stability, the CIP was moved to a freezer, maintained at 233 K, to simulate the sample loading process. This loading phase, lasting 3 to 6 minutes, was followed by transferring the CIP vessel to a hydraulic press under ambient conditions (usually at 295 K) to simulate the pressurization process, marked by the black arrow in Figure A1. Subsequently, the CIP vessel was returned to the low-temperature test chamber. The temperature of the antifreeze was recorded while meticulously adjusting the temperature in the low-temperature chamber to ensure that the temperature inside the CIP vessel equilibrated precisely with the intended annealing temperature. Through a series of such tests, we established the requisite temperature settings for various conditions, ensuring the achievement of accurate and stable annealing temperatures within one hour of loading samples.

## Appendix B: Grain-boundary identification

As mentioned in the Section 3.3, we employed Cellpose 2.0 built for CPU for segmentation of grains and identification of grain boundaries. While Cellpose 2.0 offers support for machine learning to train custom models for enhanced precision of grain-boundary identification, we found that the configured Cyto2 model performed admirably in fulfilling our requirements, although manual adjustments were required for certain samples. This version of Cellpose encountered difficulties when dealing with large images (usually > 10 Mb), compelling us to downscale the original image's dimensions by a factor of 5, as illustrated in Figure A2(a). Using Cellpose to segment grains and identify grain boundaries involves setting multiple parameters, including 'Chan to segment', 'Diameter', 'Flow threshold' and 'Cellprob threshold'. 'Chan to segment', that is, the color channel based on which the input image is to be segmented. For the sample in this experiment, the images are in grayscale, so the 'Chan to segment' parameter is set to 'Channels = [0, 0]', corresponding to grayscale. In addition, the 'Flow threshold' and 'Cellprob threshold' parameters were usually set to default values, that is, 0.4 and 0.0, respectively. For 'Diameter', which requires appropriate input based on the median estimated grain diameter (in pixels) for a given microstructure. First, the median value of the grain diameter can be automatically estimated by Cellpose and used as the input value of 'Diameter' to run Cellpose. The input value of the 'Diameter' may need to be adjusted based on the comparison between the segmentation results and the input image. After grain segmentation and grain-boundary identification by Cellpose, 'outlines' data, containing the coordinates of identified grain boundaries, can be exported to a text file. The 'outlines' data were then imported into FIJI (ImageJ) and overlaid on the input image for visualizing grain boundaries, as illustrated in Figure A2(b). The grain boundaries processed in this manner were compared with the original images, so that any mismatch boundaries (for example, uneven shades and dusts in the ethanol) were corrected manually and then converted into binary image by Adobe Photoshop 2022. An example for the

binary corrected visualized grain boundaries is illustrated in Figure A2(c). The processed grain boundaries were often wider than observed in the original images, spanning multiple pixels and potentially resulting in an underestimation of the actual grain area. To mitigate this, the grain boundaries were refined to a maximum width of two pixels using MATLAB, yielding the final grain boundaries illustrated in Figure A2(d).

### Appendix C: Bubble verification

We did not observe any bubbles in the optical micrographs of our samples, but we suspect that air bubbles, with sizes below the detection limit of our optical microscope, might have been trapped along grain boundaries in the sample (as in Section 5.3). Here we tested this hypothesis by conducting additional examinations on an ice samples doped with $10^{-2}$ mol/L KCl annealed at 258 K for 3.2 h. Initially, the sample was observed in ethanol alcohol at 213 K, as depicted in Figure A3(a). No bubbles were observed along the grain boundaries at this stage. Subsequently, we elevated the temperature of the ethanol to 233 K. After temperature stabilization, micrographs of the same region were obtained, as illustrated in Figure A3(b). Due to a higher temperature, increased ice solubility in ethanol modified the sample surface, making it difficult to directly correlate microstructures within the same area between Figure A3(a) and (b). However, at 233 K, numerous black spots appeared along the grain boundaries and continued to emerge onto the surface of the sample during the observation. The sample was then quickly cooled down to 213 K. Micrographs of the same region were taken, as illustrated in Figure A3(c). The microstructures in Figure A3(b) and (c) match well. The black spots in Figure A3(b) are air bubbles on the sample surface and in the ethanol. We interpret that these air bubbles were trapped in the ice and pressed to almost invisible during the experiment. Given the ice sample's rigidity at the lower temperature, these bubbles were immobilized and could not grow larger. At a higher temperature, the sample softened and ice dissolved more in ethanol, enabling the internal air bubbles to escape. Upon returning to the lower temperature, the bubbles became immobile again.

### Appendix D: Monte Carlo fitting

In the Methods Section we introduce how to perform the best fit. Here, we provide a more in-depth explanation of the Monte Carlo fitting process and comparisons with the outcomes from the least squares fitting. For each sample, we introduced 500 random variations to the average grain size and generated 500 grain-size data. These new grain sizes conform to a log-normal distribution with a standard deviation of 0.02. This distribution is an analogue of the actual grain-size distribution of the sample. Such that, instead of using just one average value for the fitting, a grain-size distribution was used. Examples of the generated grain-size distributions are presented in Figure A10(a). We conducted 500 fittings based on variation-added grain sizes, resulting in 500 sets of outcomes. The averages of these fittings were then calculated to obtain the final fitting results. Figure A10(b) shows histograms illustrating the results of 500 fits for parameters $d_0$, $k$, and $n$. The title of each histogram presents the mean value of the corresponding parameter. This method reduces the effect of sample variations in fitting. Figure A8 presents the $n$ values obtained through both methods, while Figure A9 illustrates a comparison between the two fitting

curves for pure ice samples at four annealing temperatures. For most samples, the disparity in values of $n$ obtained from the two methods' fitting is within 0.5. However, for samples with a value of $n$ exceeding 6 fitted by the Monte Carlo method, there is a considerable difference in the outcomes of the two methods. The least squares method yields a larger $n$ value, with the difference being at least greater than 1. Moreover, in certain samples where the 3.2-hour grain size data is lower than the initial grain size, the Monte Carlo fitting results range between 1.1 and 5.7, while the least squares fitting outcomes are typically much greater than 20. However, these samples also exhibit substantial grain growth over longer annealing times, indicating that such large $n$ values are clearly inconsistent with the grain size data. This underscores that, relative to traditional least squares method fitting, the Monte Carlo method can minimize the impact of the sample variations. Compared to the conventional least squares fitting method that relies solely on the average grain size, the Monte Carlo method does not change the original data, but incorporates the grain-size distribution into the fitting process, enhancing the reliability of the fit.

## Appendix E:  Supplementary table and figures

- Figure A1
- Figure A2
- Figure A3
- Figure A4
- Figure A5
- Figure A6
- Figure A7
- Figure A8
- Figure A9
- Figure A10
- Table A1

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

**Table 1.** Grain-size data for all samples.

| Temperature (K) | 268 | | | | | | | 263 | | | | | | |
|---|---|---|---|---|---|---|---|---|---|---|---|---|---|---|
| Composition | KCl | | | | MgSO$_4$ | | pure ice | KCl | | | | MgSO$_4$ | | pure ice |
| Concentration (mol/L) | $10^{-2}$ | $10^{-3}$ | $10^{-4}$ | $10^{-5}$ | $10^{-2}$ | $10^{-5}$ | | $10^{-2}$ | $10^{-3}$ | $10^{-4}$ | $10^{-5}$ | $10^{-2}$ | $10^{-5}$ | |
| $d$ (μm) 0 h[a] | 46 | 49 | 43 | 48 | 42 | 47 | 55 | 46 | 49 | 43 | 48 | 42 | 47 | 55 |
| 3.2 h | 104 | 107 | 121 | 117 | 70 | 60 | 74 | 80 | 83 | 80 | 79 | 64 | 76 | 78 |
| 10 h | 123 | 127 | 155 | 153 | 120 | 83 | 110 | 92 | 113 | 96 | 86 | 68 | 86 | 109 |
| 32 h | 184 | 180 | 205 | 218 | 166 | 138 | 138 | 112 | 133 | 117 | 98 | 83 | 94 | 133 |
| 100 h | 223 | 225 | 226 | 315 | 192 | 177 | 206 | 167 | 168 | 149 | 138 | 113 | 125 | 162 |
| $\sigma_d/d$ 0h | 0.44 | 0.43 | 0.41 | 0.46 | 0.55 | 0.55 | 0.43 | 0.44 | 0.43 | 0.41 | 0.46 | 0.55 | 0.55 | 0.43 |
| 3.2h | 0.48 | 0.45 | 0.47 | 0.55 | 0.46 | 0.48 | 0.41 | 0.42 | 0.49 | 0.48 | 0.42 | 0.41 | 0.44 | 0.41 |
| 10h | 0.48 | 0.56 | 0.57 | 0.76 | 0.48 | 0.54 | 0.40 | 0.61 | 0.57 | 0.45 | 0.43 | 0.45 | 0.43 | 0.48 |
| 32h | 0.65 | 0.71 | 0.56 | 0.60 | 0.52 | 0.55 | 0.65 | 0.58 | 0.58 | 0.44 | 0.55 | 0.50 | 0.52 | 0.54 |
| 100h | 0.58 | 0.61 | 0.79 | 0.59 | 0.55 | 0.59 | 0.86 | 0.61 | 0.58 | 0.72 | 0.75 | 0.55 | 0.51 | 0.55 |
| $n$ | 4.2±0.4 | 4.4±0.4 | 5.1±0.5 | 3.4±0.2 | 3.4±0.2 | 2.6±0.2 | 3.1±0.3 | 4.6±0.5 | 5.0±0.5 | 5.5±0.5 | 5.9±0.5 | 5.5±0.6 | 6.1±0.3 | 4.6±0.4 |
| $k$ (m$^4$/s) fit with $n=4$ | $8.0\times10^{-21}$ | $8.4\times10^{-21}$ | $1.4\times10^{-20}$ | $1.9\times10^{-20}$ | $4.1\times10^{-21}$ | $1.8\times10^{-21}$ | $3.4\times10^{-21}$ | $1.9\times10^{-21}$ | $3.1\times10^{-21}$ | $1.9\times10^{-21}$ | $1.2\times10^{-21}$ | $5.0\times10^{-21}$ | $9.3\times10^{-22}$ | $2.5\times10^{-21}$ |
| $R^2$ | 0.99 | 0.99 | 0.98 | 0.99 | 0.96 | 0.95 | 0.97 | 0.97 | 0.98 | 0.97 | 0.92 | 0.94 | 0.89 | 0.97 |

| Temperature (K) | 258 | | | | | | | 253 | | | | | | |
|---|---|---|---|---|---|---|---|---|---|---|---|---|---|---|
| Composition | KCl | | | | MgSO$_4$ | | pure ice | KCl | | | | MgSO$_4$ | | pure ice |
| Concentration (mol/L) | $10^{-2}$ | $10^{-3}$ | $10^{-4}$ | $10^{-5}$ | $10^{-2}$ | $10^{-5}$ | | $10^{-2}$ | $10^{-3}$ | $10^{-4}$ | $10^{-5}$ | $10^{-2}$ | $10^{-5}$ | |
| $d$ (μm) 0 h | 46 | 49 | 43 | 48 | 42 | 47 | 55 | 46 | 49 | 43 | 48 | 42 | 47 | 55 |
| 3.2 h | 60 | 64 | 70 | 72 | 44 | 44 | 62 | 36 | 51 | 65 | 70 | 40 | 60 | 51 |
| 10 h | 67 | 86 | 92 | 75 | 56 | 46 | 82 | 47 | 58 | 72 | 90 | 46 | 63 | 84 |
| 32 h | 77 | 101 | 102 | 87 | 58 | 54 | 106 | 65 | 68 | 98 | 102 | 61 | 65 | 97 |
| 100 h | 107 | 117 | 109 | 111 | 80 | 81 | 134 | 80 | 88 | 109 | 120 | 68 | 81 | 122 |
| $\sigma_d/d$ 0h | 0.44 | 0.43 | 0.41 | 0.46 | 0.55 | 0.55 | 0.43 | 0.44 | 0.43 | 0.41 | 0.46 | 0.55 | 0.55 | 0.43 |
| 3.2h | 0.43 | 0.39 | 0.47 | 0.44 | 0.45 | 0.43 | 0.49 | 0.45 | 0.39 | 0.45 | 0.52 | 0.43 | 0.43 | 0.46 |
| 10h | 0.43 | 0.37 | 0.41 | 0.51 | 0.38 | 0.44 | 0.41 | 0.48 | 0.44 | 0.53 | 0.49 | 0.40 | 0.44 | 0.50 |
| 32h | 0.45 | 0.54 | 0.54 | 0.46 | 0.42 | 0.48 | 0.48 | 0.47 | 0.38 | 0.55 | 0.51 | 0.50 | 0.43 | 0.50 |
| 100h | 0.43 | 0.40 | 0.62 | 0.55 | 0.51 | 0.51 | 0.51 | 0.44 | 0.38 | 0.45 | 0.60 | 0.43 | 0.55 | 0.49 |
| $n$ | 4.9±1.0 | 5.6±0.5 | 6.2±0.1 | 6.1±0.2 | 4.1±1.4 | 1.1±0.4 | 3.8±0.5 | 2.7±0.5 | 3.6±1.2 | 5.7±0.5 | 6.0±0.3 | 4.4±0.8 | 5.7±1.0 | 3.7±0.5 |
| $k$ (m$^4$/s) fit with $n=4$ | $3.5\times10^{-22}$ | $7.9\times10^{-22}$ | $9.5\times10^{-22}$ | $5.2\times10^{-22}$ | $9.8\times10^{-23}$ | $7.1\times10^{-23}$ | $9.1\times10^{-22}$ | $1.9\times10^{-21}$ | $1.4\times10^{-22}$ | $6.3\times10^{-22}$ | $9.7\times10^{-22}$ | $6.4\times10^{-23}$ | $1.1\times10^{-22}$ | $6.4\times10^{-22}$ |
| $R^2$ | 0.95 | 0.93 | 0.83 | 0.87 | 0.93 | 0.87 | 0.98 | 0.82 | 0.97 | 0.93 | 0.91 | 0.91 | 0.78 | 0.89 |

[a]The starting materials, i.e. 0-h samples, are the same for different temperatures.

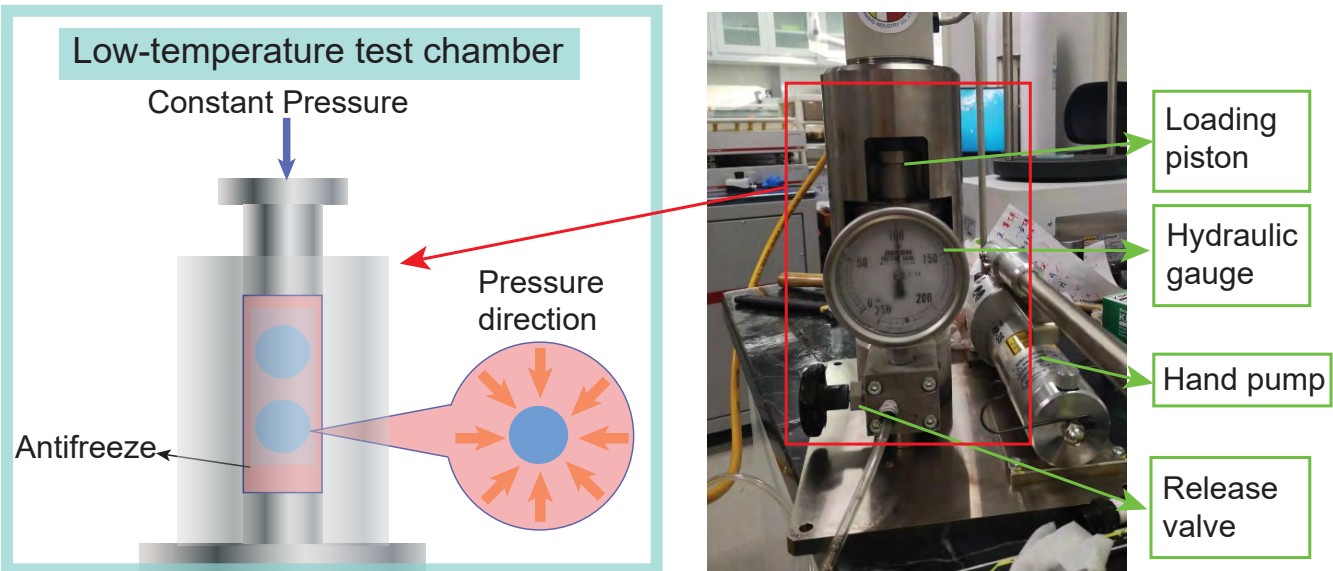

**Figure 1.** Photo and drawing for the cold isostatic press (CIP). On the right is a photo showing CIP under a pressure of 100 MPa. On the left is a drawing of the pressure vessel in the low-temperature test chamber, which samples inside. Antifreeze, acting as the pressure medium, delivers hydrostatic pressure on the samples.

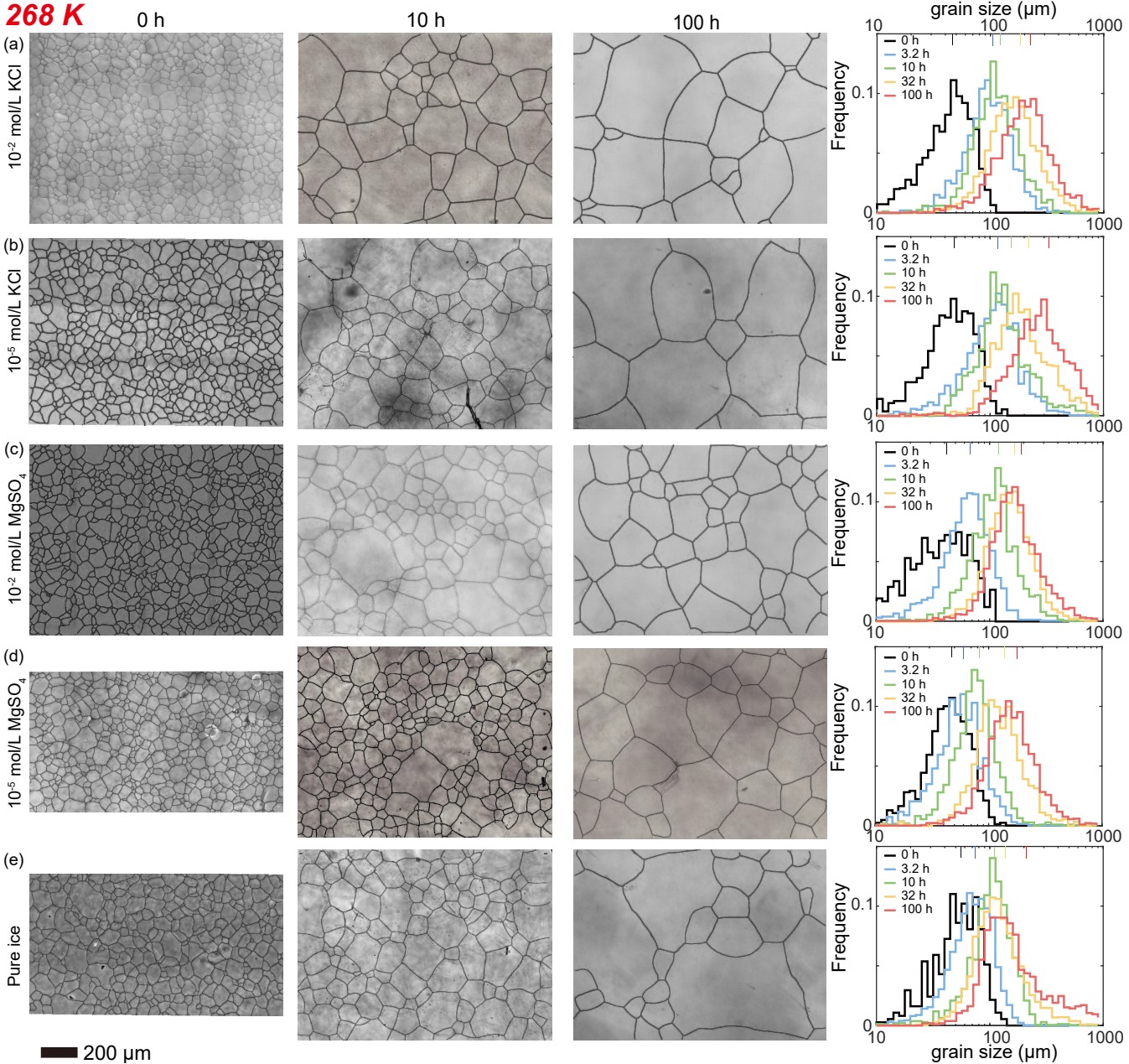

**Figure 2.** Optical images of microstructure and histograms of grain-size distribution of the highest and lowest ($10^{-2}$ and $10^{-5}$ mol/L, respectively) concentrations of doped ice samples as well as pure water ice samples annealed at 268 K, the highest annealing temperature. (a) Ice + $10^{-2}$ mol/L KCl. (b) Ice + $10^{-5}$ mol/L KCl. (c) Ice + $10^{-2}$ mol/L $MgSO_4$. (d) Ice + $10^{-5}$ mol/L $MgSO_4$. (e) Pure ice. In each panel, the micrograph of the starting sample is on the first column, the micrographs of samples annealed to 10 and 100 h are on the second and third columns, respectively, and a plot of histograms of grain-size distributions for all 5 samples for this impurity concentration is on the fourth column. Note that annealing experiments at different temperatures use the same 0-h samples. The micrographs were processed from original ones after shading corrections and contrast adjustments. Scale bar, presented at the bottom, applies for all panels. In the plot of histograms, colored lines on the top represent the arithmetic mean of the grain size, with each color corresponding to an annealing time. For results of samples doped with intermediate concentrations of KCl, please see Appendix Figures A4 and A5

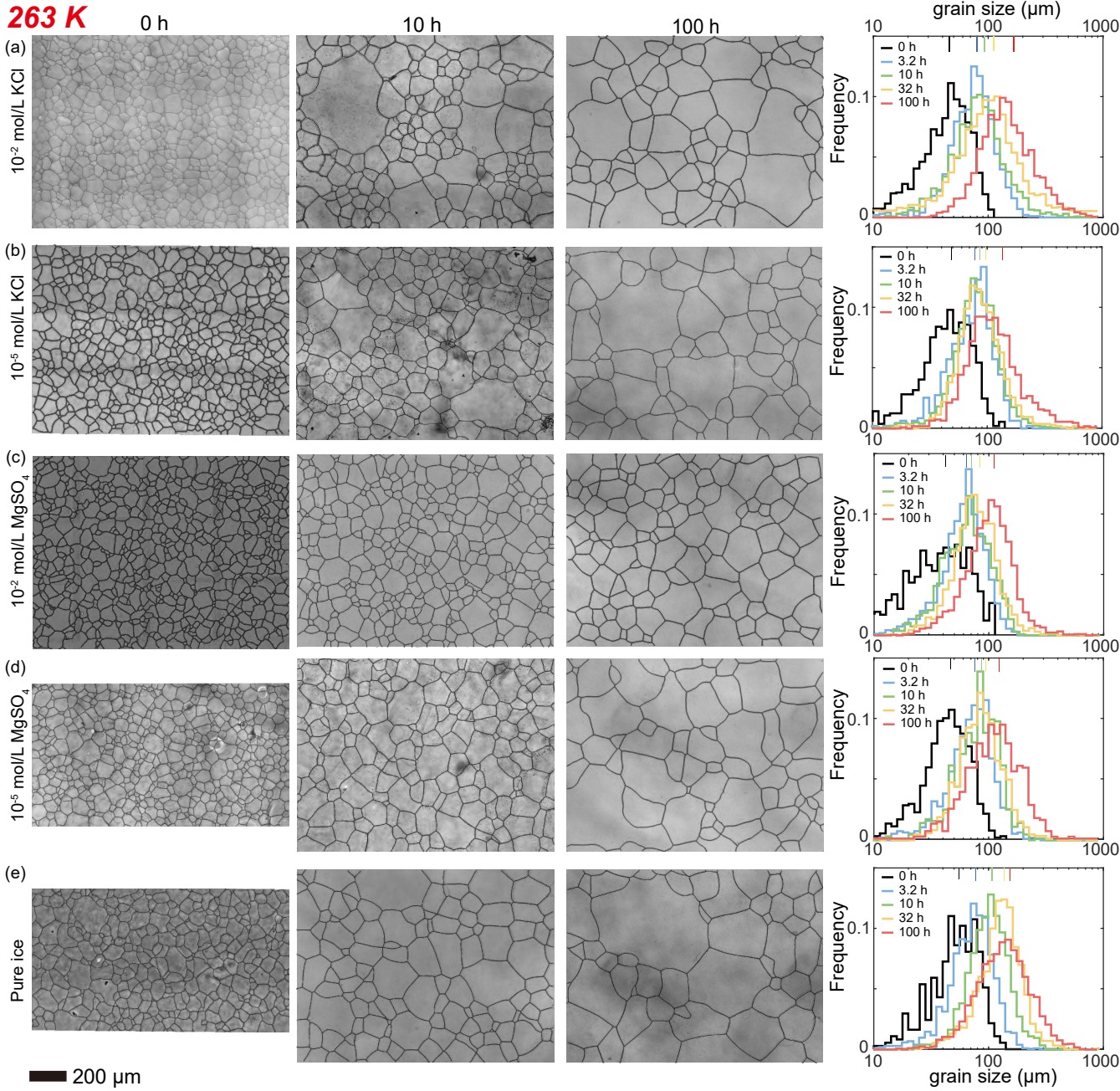

**Figure 3.** Optical images of microstructure and histograms of grain-size distribution of the highest and lowest ($10^{-2}$ and $10^{-5}$ mol/L, respectively) concentrations of doped ice samples as well as pure water ice samples annealed at 263 K. (a) Ice + $10^{-2}$ mol/L KCl. (b) Ice + $10^{-5}$ mol/L KCl. (c) Ice + $10^{-2}$ mol/L MgSO$_4$. (d) Ice + $10^{-5}$ mol/L MgSO$_4$. (e) Pure ice. In each panel, the micrograph of the starting sample is on the first column, the micrographs of samples annealed to 10 and 100 h are on the second and third columns, respectively, and a plot of histograms of grain-size distributions for all 5 samples for this impurity concentration is on the fourth column. Note that annealing experiments at different temperatures use the same 0-h samples. Scale bar, presented at the bottom, applies for all panels. In the plot of histograms, colored lines on the top represent the arithmetic mean of the grain size, with each color corresponding to an annealing time.

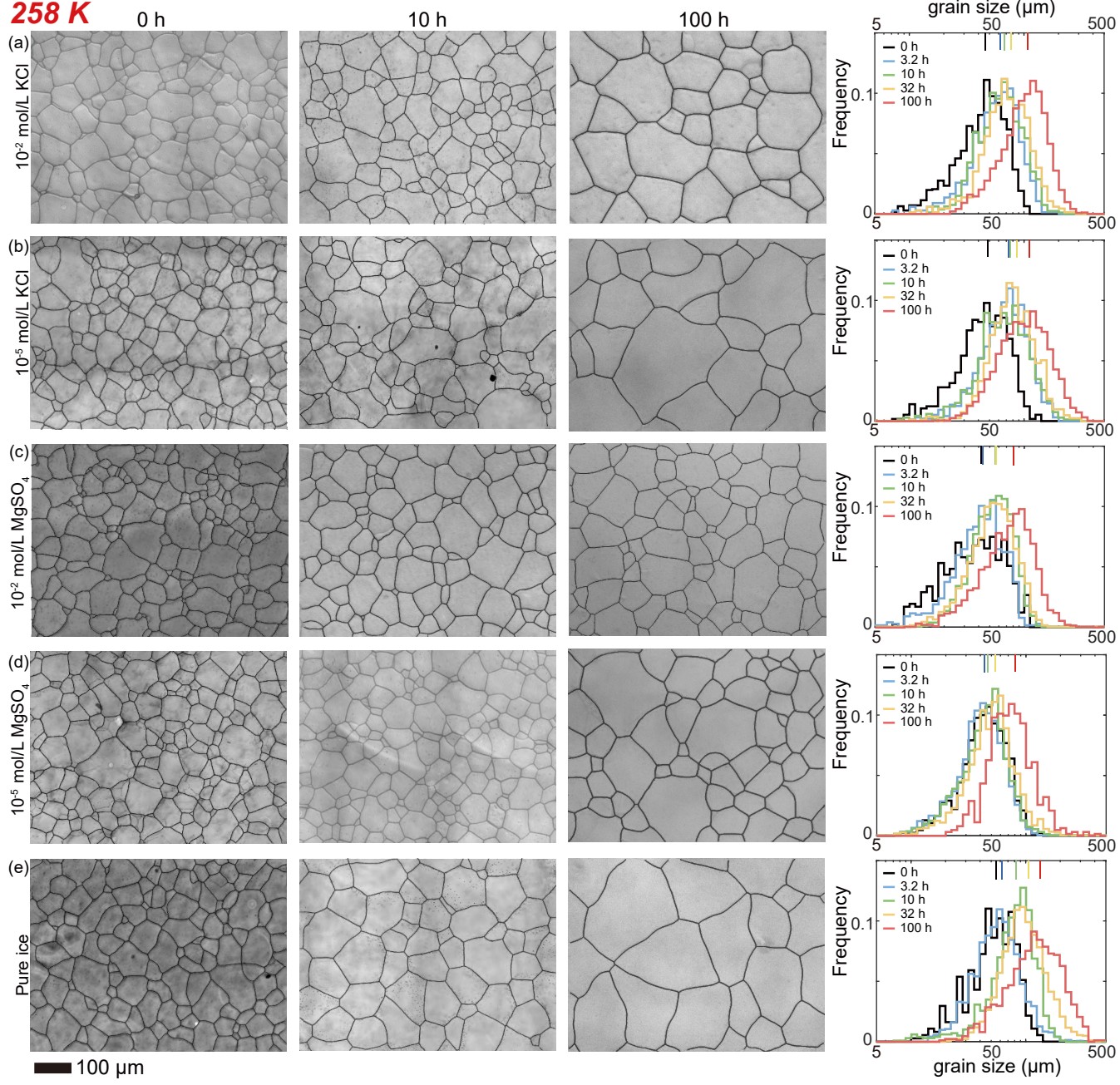

**Figure 4.** Optical images of microstructure and histograms of grain-size distribution of the highest and lowest ($10^{-2}$ and $10^{-5}$ mol/L, respectively) concentrations of doped ice samples as well as pure water ice samples annealed at 258 K. (a) Ice + $10^{-2}$ mol/L KCl. (b) Ice + $10^{-5}$ mol/L KCl. (c) Ice + $10^{-2}$ mol/L MgSO$_4$. (d) Ice + $10^{-5}$ mol/L MgSO$_4$. (e) Pure ice. In each panel, the micrograph of the starting sample is on the first column, the micrographs of samples annealed to 10 and 100 h are on the second and third columns, respectively, and a plot of histograms of grain-size distributions for all 5 samples for this impurity concentration is on the fourth column. Note that annealing experiments at different temperatures use the same 0-h samples. Scale bar, presented at the bottom, applies for all panels. In the plot of histograms, colored lines on the top represent the arithmetic mean of the grain size, with each color corresponding to an annealing time.

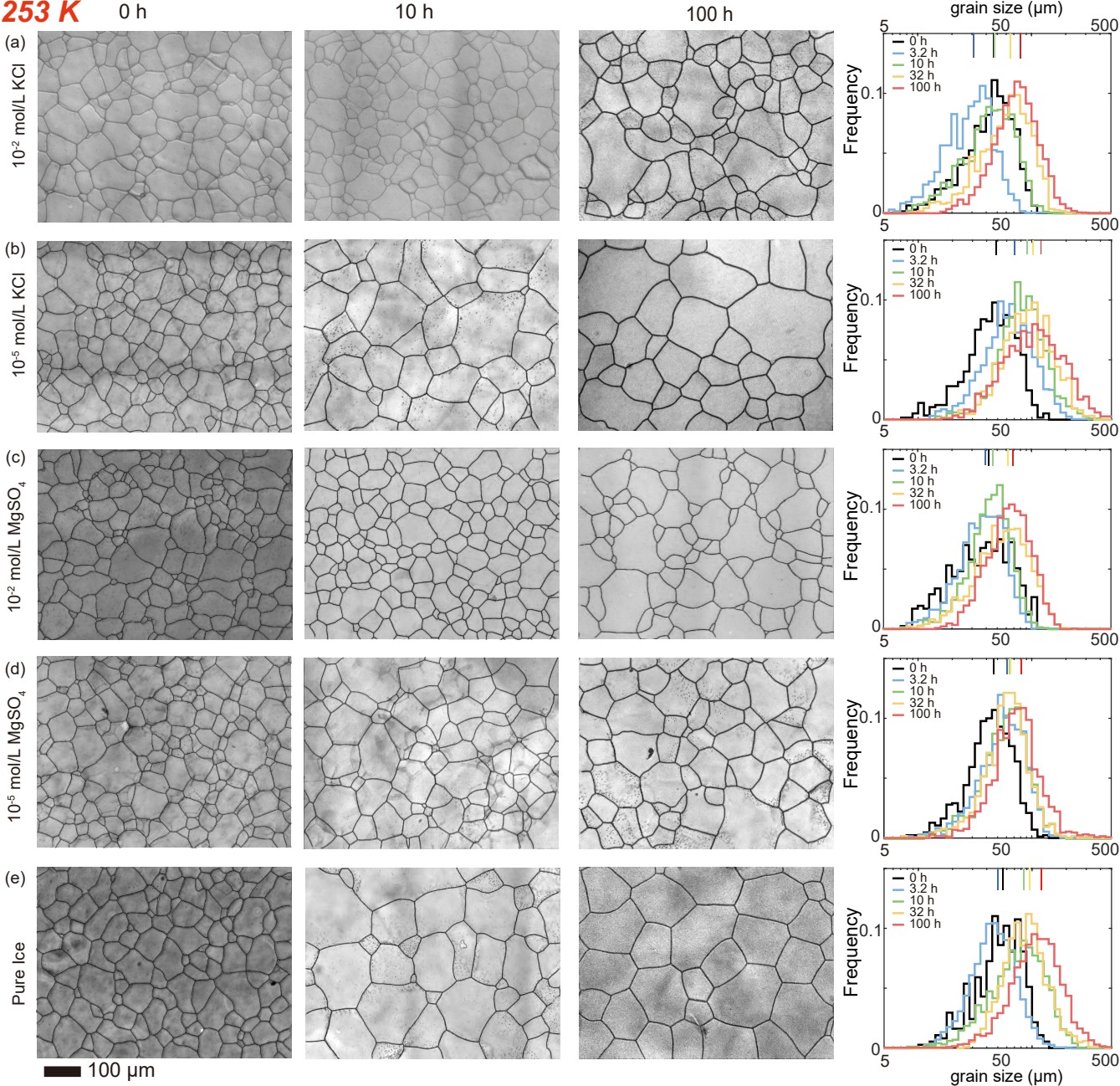

**Figure 5.** Optical images of microstructure and histograms of grain-size distribution of the highest and lowest ($10^{-2}$ and $10^{-5}$ mol/L, respectively) concentrations of doped ice samples as well as pure water ice samples annealed at 253 K, the lowest annealing temperature. (a) Ice + $10^{-2}$ mol/L KCl. (b) Ice + $10^{-5}$ mol/L KCl. (c) Ice + $10^{-2}$ mol/L MgSO$_4$. (d) Ice + $10^{-5}$ mol/L MgSO$_4$. (e) Pure ice. In each panel, the micrograph of the starting sample is on the first column, the micrographs of samples annealed to 10 and 100 h are on the second and third columns, respectively, and a plot of histograms of grain-size distributions for all 5 samples for this impurity concentration is on the fourth column. Note that annealing experiments at different temperatures use the same 0-h samples. Scale bar, presented at the bottom, applies for all panels. In the plot of histograms, colored lines on the top represent the arithmetic mean of the grain size, with each color corresponding to an annealing time. The little black dots in panels (b), (d) and (e) are bubbles in the alcohol, which were not completely removed during the observation of these samples.

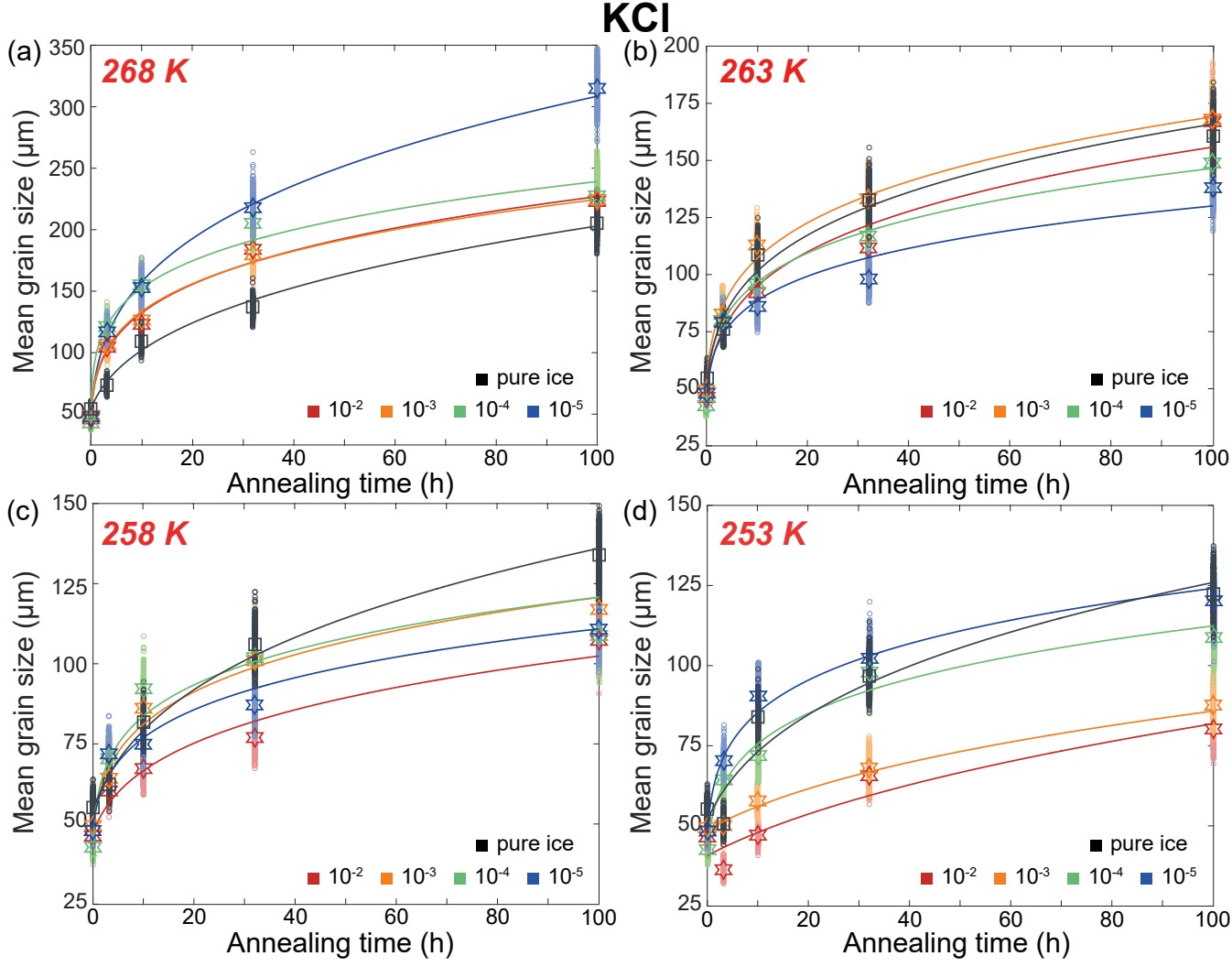

**Figure 6.** Plots of grain size versus annealing time for samples of KCl-doped ice and pure ice at (a) 268 K, (b) 263 K, (c) 258 K and (d) 253 K. Different colors represent different concentrations of KCl, with the unit being mol/L. Black represents pure-water-ice samples. The stars (KCl-doped ice) and squares (pure ice) are the measured average grain size, and small circles are the data sets with noise added. The solid curves are the best fits.

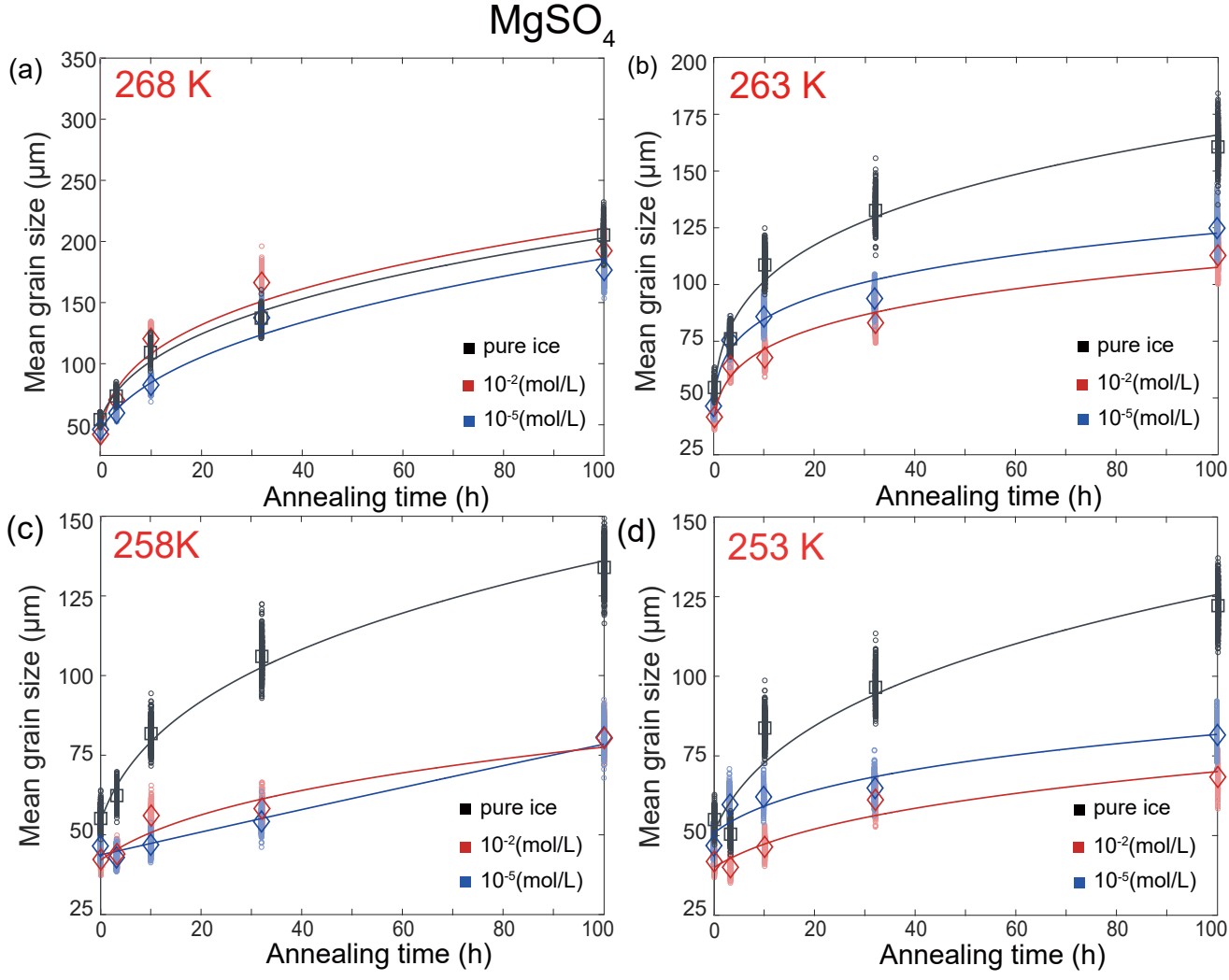

**Figure 7.** Plots of grain size versus annealing time for samples of MgSO$_4$-doped ice and pure ice at (a) 268 K, (b) 263 K, (c) 258 K and (d) 253 K. Different colors represent different concentrations of KCl, with the unit being mol/L. Black represents pure-water-ice samples. The diamonds (MgSO$_4$-doped ice) and squares (pure ice) are the measured average grain size, and small circles are the data sets with noise added. The solid curves are the best fits.

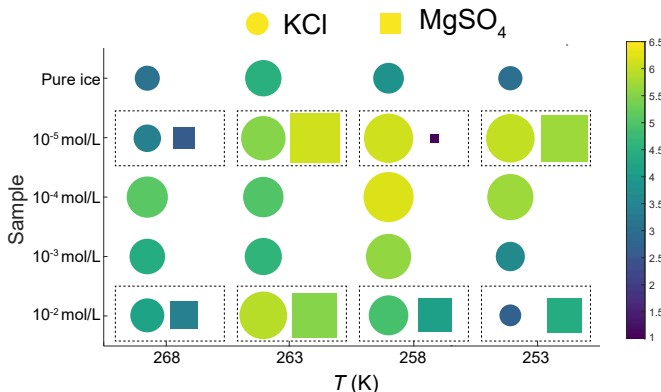

**Figure 8.** Plots of grain size exponent, $n$, versus temperature and sample type. Round symbols are KCl-doped ice, and square symbols are MgSO4-doped ice. The symbols within the dashed rectangle represent ice doped with the same concentration and annealed at the same temperature. Symbols are colored with respect to the value of $n$. Larger size and warmer color represent a larger value of $n$.

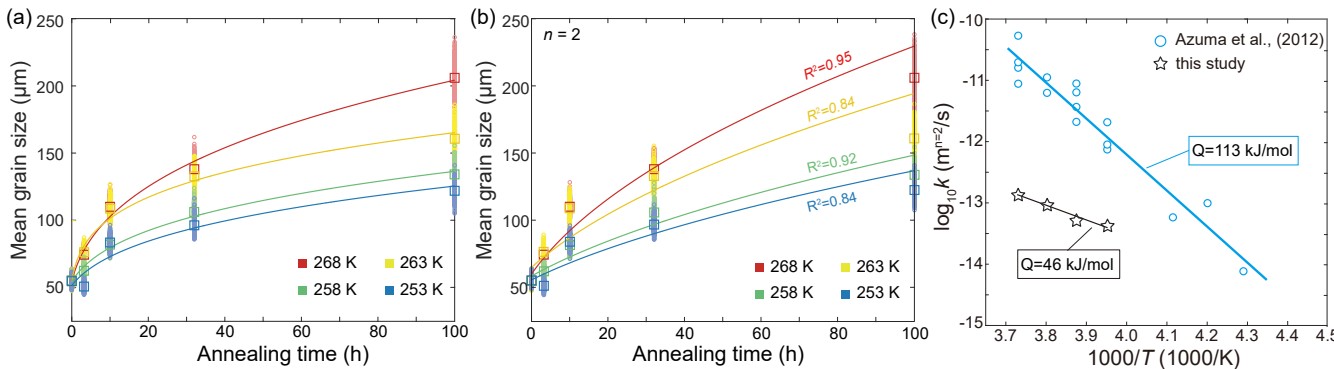

**Figure 9.** (a) Plots of grain size versus annealing time for samples of pure ice. Different colors represent different temperatures. The squares are the measured average grain size, and small circles are the data sets with noise added. The solid curves are the best fits. (b) Plots of grain size versus annealing time for samples of pure ice with $n$ fixed to 2. The goodness of fits ($R^2$) are presented next to the curves, respectively. (c) Comparison of the grain growth rate constant $k$ of bubble-free pure ice from Azuma et al. (2012) and this study, calculated by fixing $n = 2$.

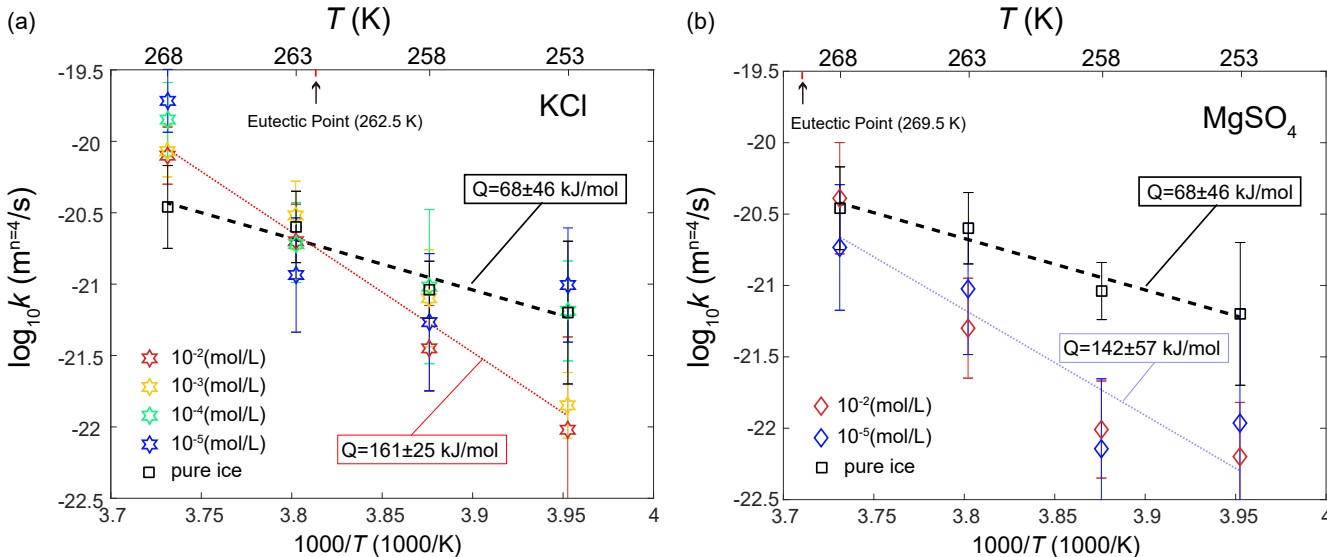

**Figure 10.** Temperature dependence of the grain-growth rate constant $k$ of (a) KCl-doped ice and (b) MgSO$_4$-doped ice calculated by fixing $n = 4$. Data for pure ice were plotted as black squares in both panels as a reference. Different colors represent different impurity concentrations. The errors of $k$ are indicated by the vertical lines. The black dashed line is the fit of pure ice, while the colored dotted lines are the fits for doped ice.

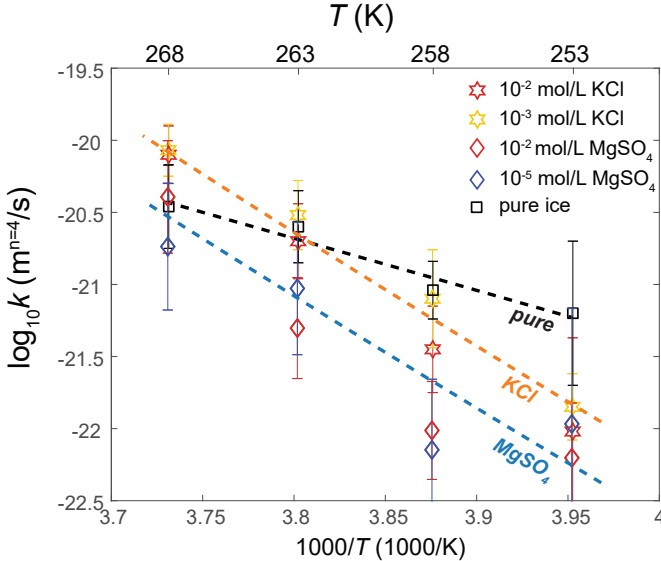

**Figure 11.** Temperature dependence of the grain-growth rate constant $k$ of doped ice and pure ice calculated by fixing $n = 4$. Samples that exhibit an effect on grain growth at temperatures both above and below the eutectic point are plotted. Stars represent KCl-doped ice and diamonds represent $MgSO_4$-doped ice. Different colors represent different impurity concentrations. The errors of $k$ are indicated by the vertical lines. The black dashed line is the fit for pure ice. The orange and blue dashed lines are the fits based on Eq. 7 for KCl-doped and $MgSO_4$-doped ices, respectively.

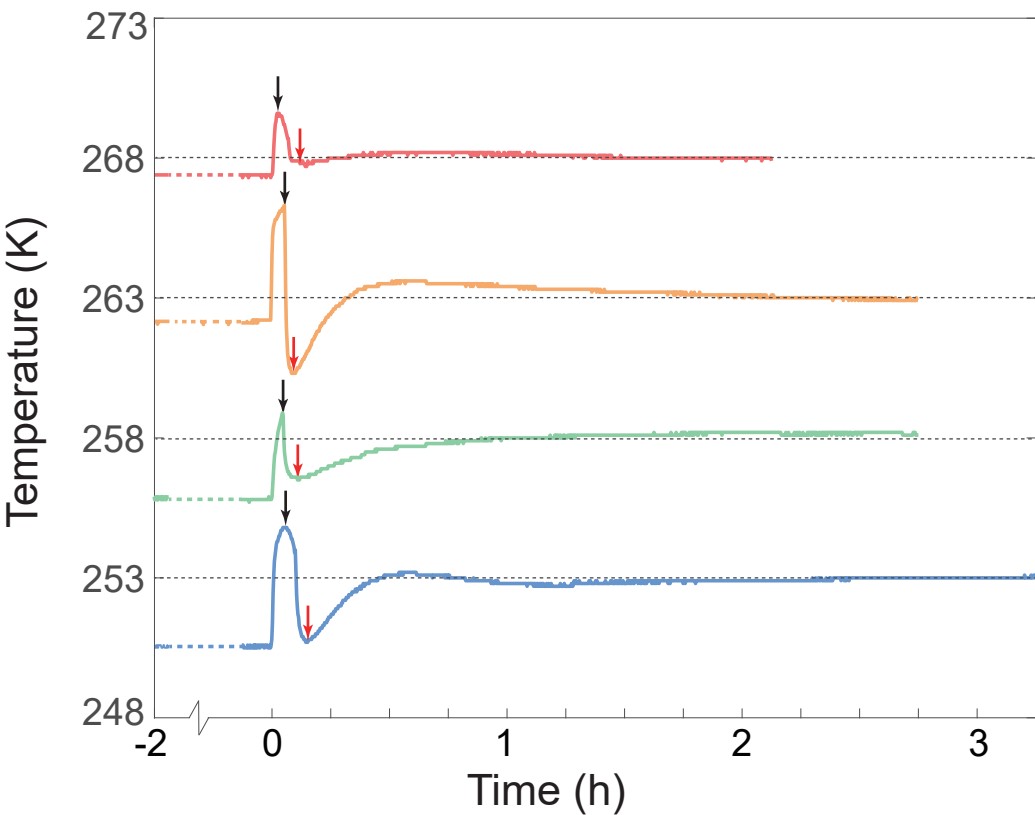

**Figure A1.** Temperature calibration in non-pressurized CIP vessel. The black dotted line represents the target temperature. Each colored curve represents the temperature of the antifreeze in the pressure vessel during the calibration experiment. Before starting an experiment, the antifreeze was pre-cooled in the vessel of the CIP in the low-temperature test chamber for at least 2 h. The pre-cooled temperature is indicated by a colored dotted line before 0 h. The rise in temperature at 0 hours is due to the removal of the thermocouple from the vessel. This step corresponds to the pressurization process. The black arrow indicates when the thermocouple was re-immersed in the antifreeze. This step corresponds to returning the pressurized CIP to the low-temperature test chamber. The red arrow points to the lowest temperature during the temperature re-stabilization process in the low-temperature test chamber. The temperature of the antifreeze in the pressure vessel usually took less than 1 h to become stable at the target temperature.

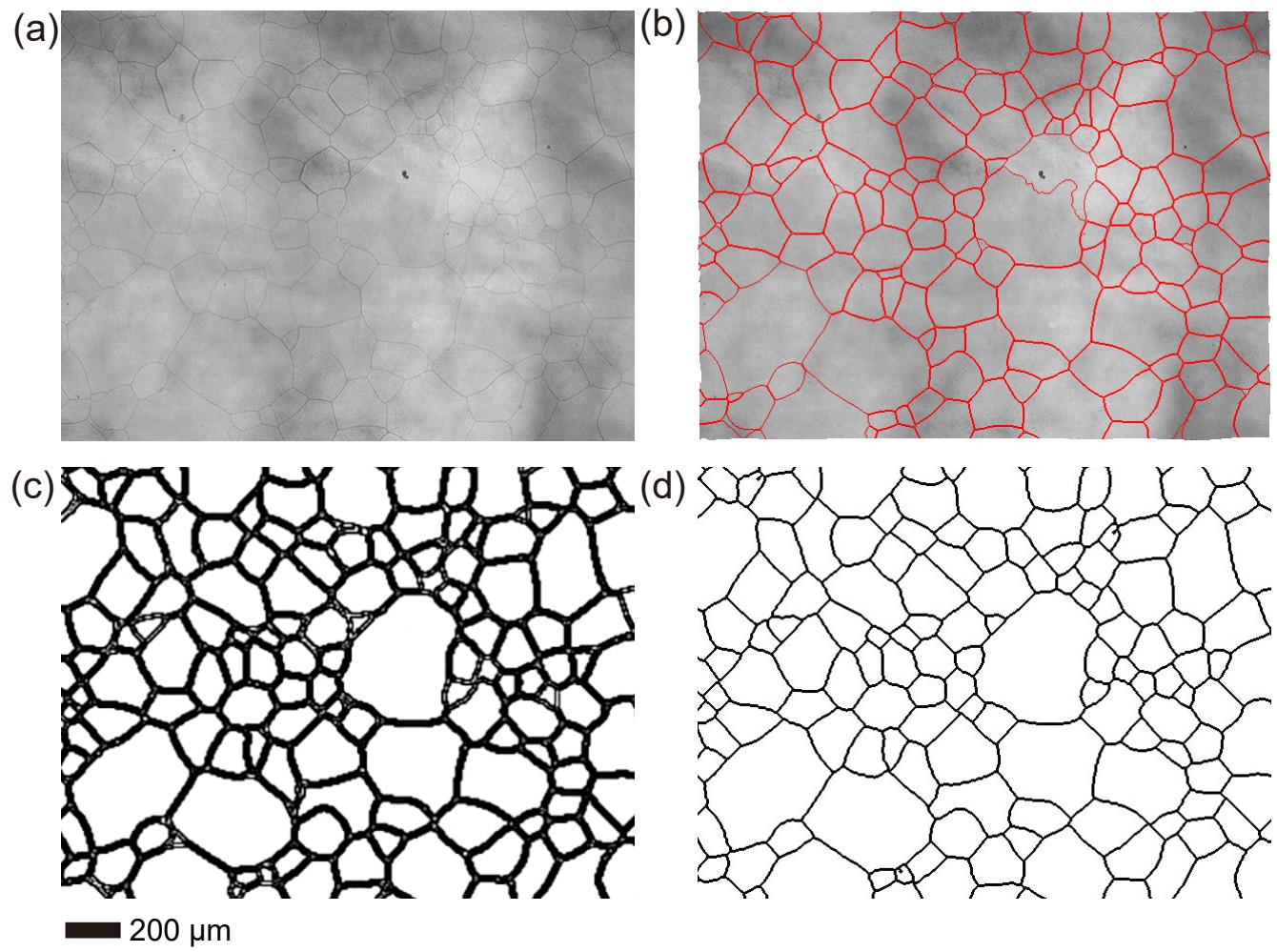

**Figure A2.** Identification, correction and refinement of grain boundaries. (a) The input image. (b) Visualized grain boundaries overlaid on the input image. (c) Grain boundaries subsequent to manual correction and conversion to a binary format. Note that the curvy "grain boundary" in the large grain in the center of the image was manually removed. (d) Refined grain boundaries with a maximum width of two pixels.

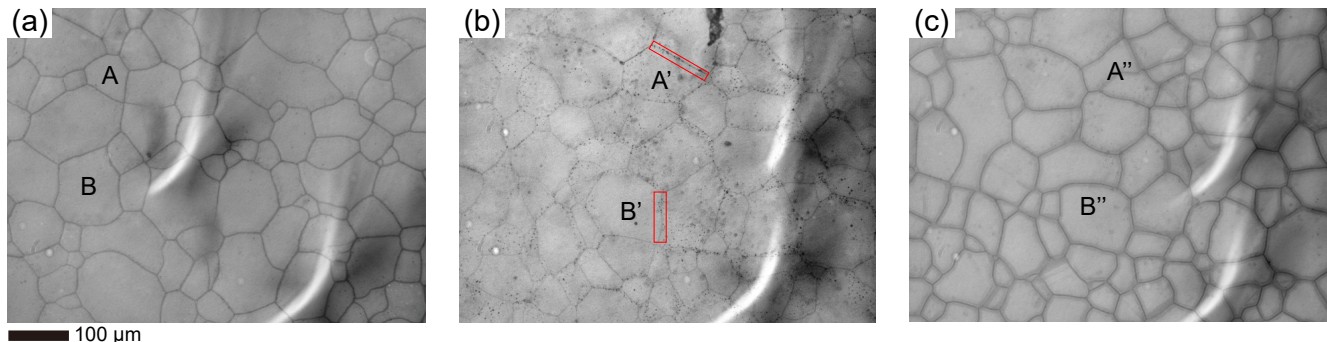

**Figure A3.** Verification of the presence of air bubbles in the sample. An ice sample doped with $10^{-2}$ mol/L KCl and annealed at 258 K for 3.2 hours was used as an example. (a) Microstructure of the surface of the sample observed at 213 K. (b) Microstructure of the same area as (a), with the temperature increased to 233 K. The red rectangular highlights bubbles along the grain boundary. (c) Microstructure of the same area as (a) and (b), with the temperature decreased back to 213 K. In the figure, A, A', and A" (B, B', and B") should represent the same grain.

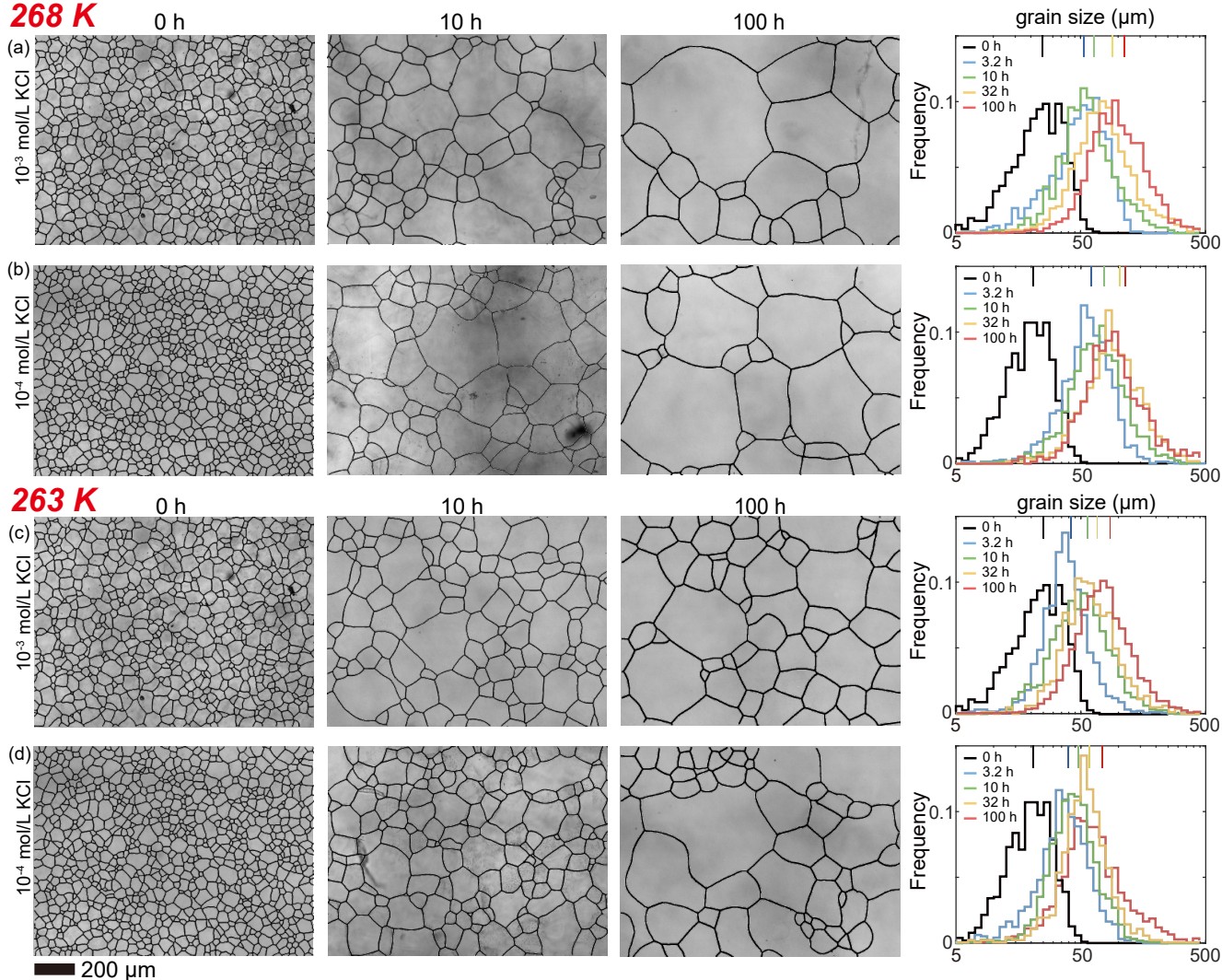

**Figure A4.** Optical images of microstructure and histograms of grain-size distribution of the medium ($10^{-3}$ and $10^{-4}$ mol/L, respectively) concentrations of KCl-doped ice samples annealed at 268 K and 263 K. (a) Ice + $10^{-3}$ mol/L KCl annealed at 268 K. (b) Ice + $10^{-4}$ mol/L KCl annealed at 268 K. (c) Ice + $10^{-3}$ mol/L KCl annealed at 263 K. (d) Ice + $10^{-4}$ mol/L KCl annealed at 263 K. In each panel, the micrograph of the starting sample is on the first column, the micrographs of samples annealed to 10 and 100 h are on the second and third columns, respectively, and a plot of histograms of grain-size distributions for all 5 samples for this impurity concentration is on the fourth column. Note that annealing experiments at different temperatures use the same 0-h samples. Scale bar, presented at the bottom, applies for all panels. In the plot of histograms, colored lines on the top represent the arithmetic mean of the grain size, with each color corresponding to an annealing time.

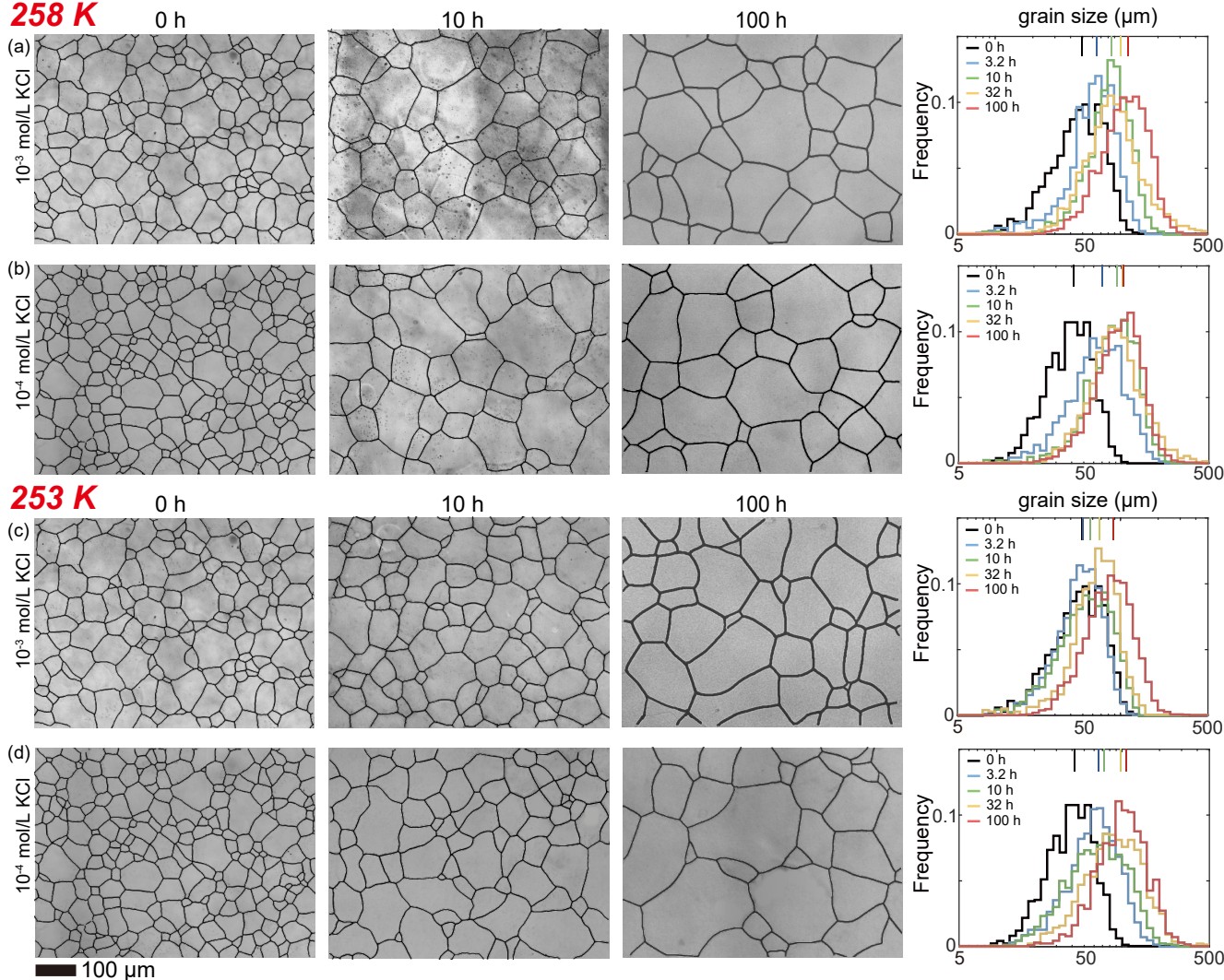

**Figure A5.** Optical images of microstructure and histograms of grain-size distribution of the medium ($10^{-3}$ and $10^{-4}$ mol/L, respectively) concentrations of KCl-doped ice samples annealed at 258 K and 253 K. (a) Ice + $10^{-3}$ mol/L KCl annealed at 258 K. (b) Ice + $10^{-4}$ mol/L KCl annealed at 258 K. (c) Ice + $10^{-3}$ mol/L KCl annealed at 253 K. (d) Ice + $10^{-4}$ mol/L KCl annealed at 253 K. In each panel, the micrograph of the starting sample is on the first column, the micrographs of samples annealed to 10 and 100 h are on the second and third columns, respectively, and a plot of histograms of grain-size distributions for all 5 samples for this impurity concentration is on the fourth column. Note that annealing experiments at different temperatures use the same 0-h samples. Scale bar, presented at the bottom, applies for all panels. In the plot of histograms, colored lines on the top represent the arithmetic mean of the grain size, with each color corresponding to an annealing time.

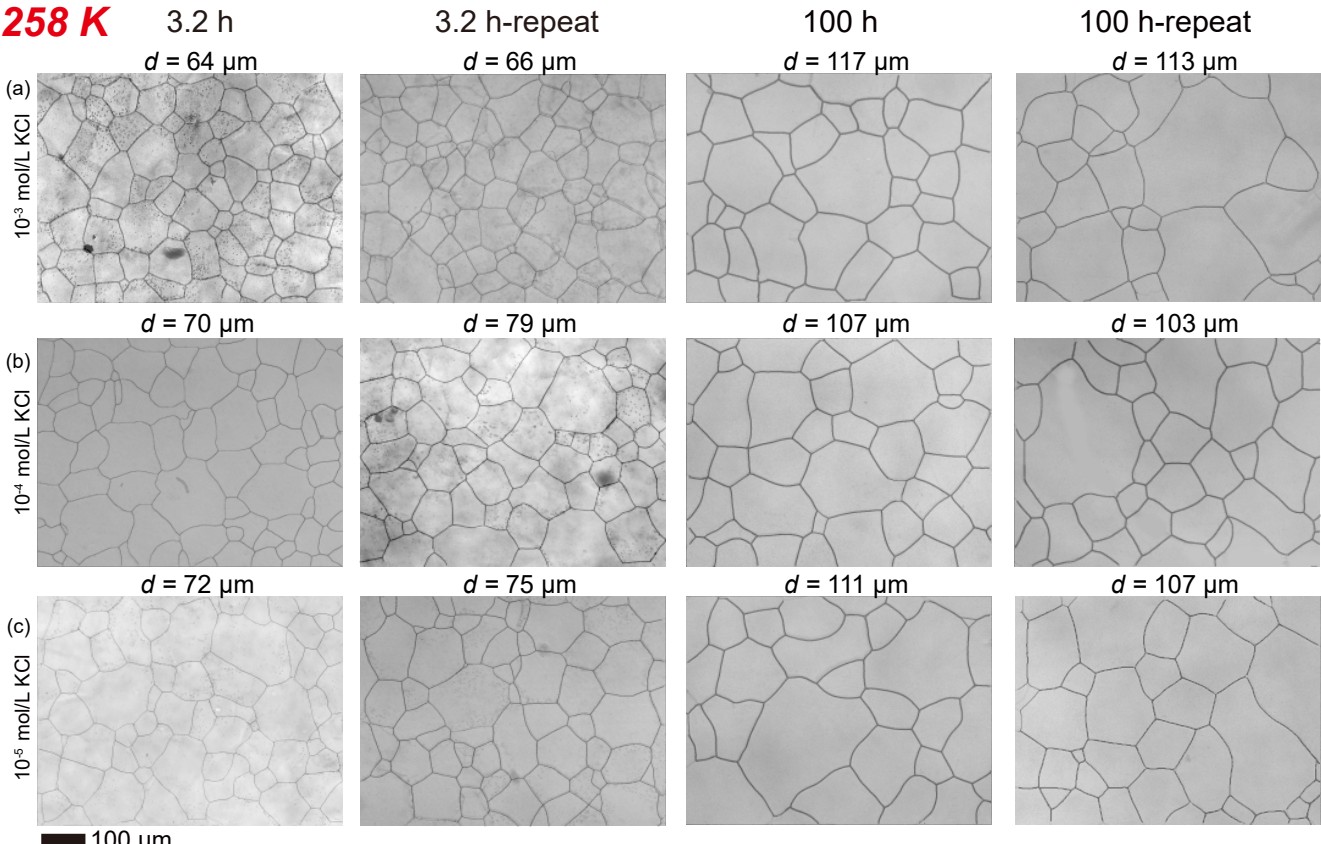

**Figure A6.** Optical images of microstructure of two groups of samples annealed at the same conditions for reproducibility check. The samples are ice doped with (a) $10^{-3}$ mol/L, (b) $10^{-4}$ mol/L and (c) $10^{-5}$ mol/L of KCl, respectively. In each panel, the first and third columns are the micrographs for the samples annealed for 3.2 h and 100 h (samples used in the manuscript), respectively, and the second and fourth columns are the micrographs of the second group of samples annealed for the same duration (marked as "-repeat"), respectively. The average grain diameter of each sample is given on the top of each micrograph. The scale bar is shown at the bottom and applies to all panels.

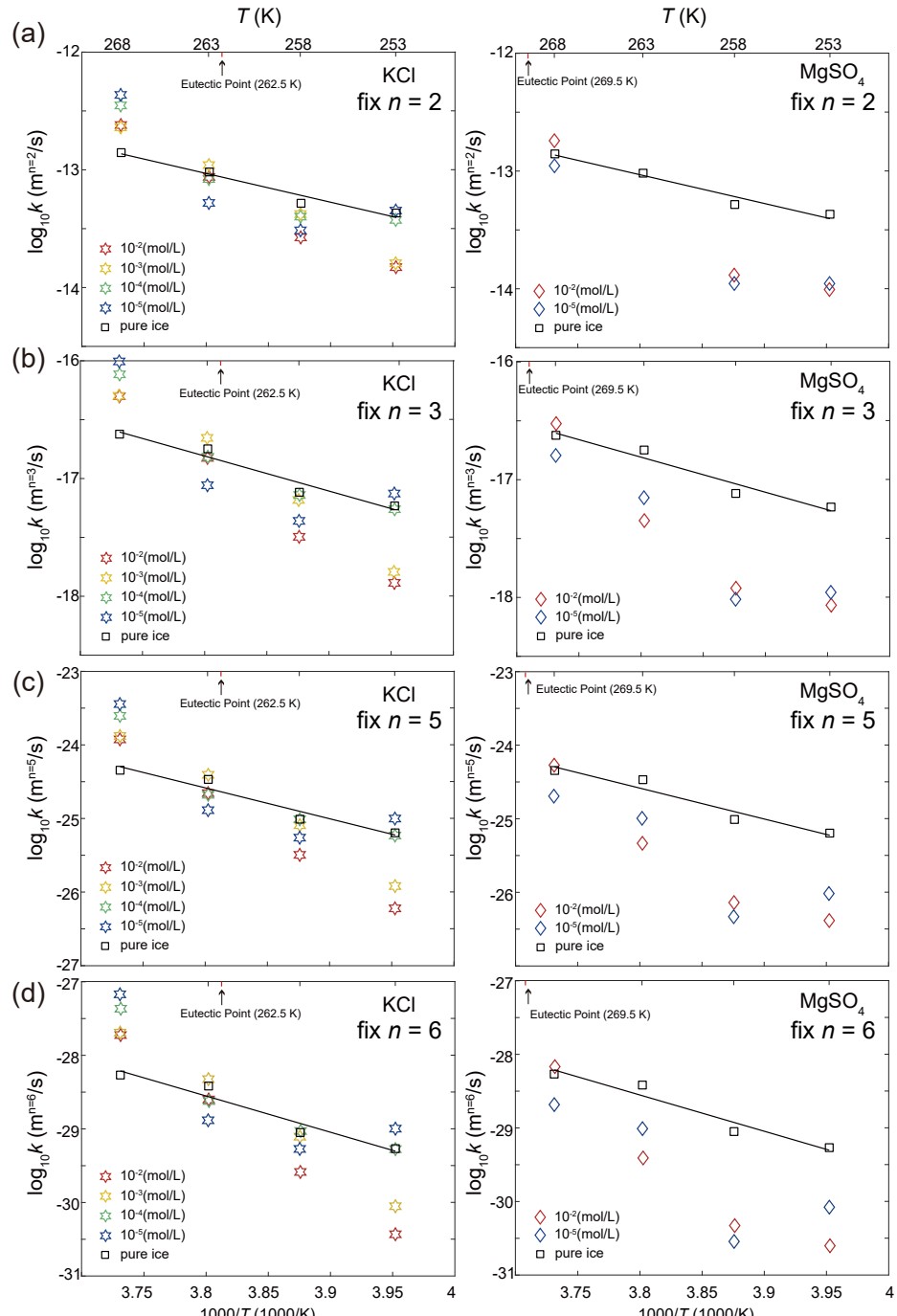

**Figure A7.** Temperature dependence of the grain-growth rate constant $k$ of KCl-doped ice (left column) and $MgSO_4$-doped ice (right column) calculated by fixing different values of $n$. (a) $n = 2$, (b) $n = 3$, (c) $n = 5$, (d) $n = 6$. Data for pure ice were plotted as black squares in both panels as a reference. Different colors represent different impurity concentrations. The black line is the fit of pure ice.

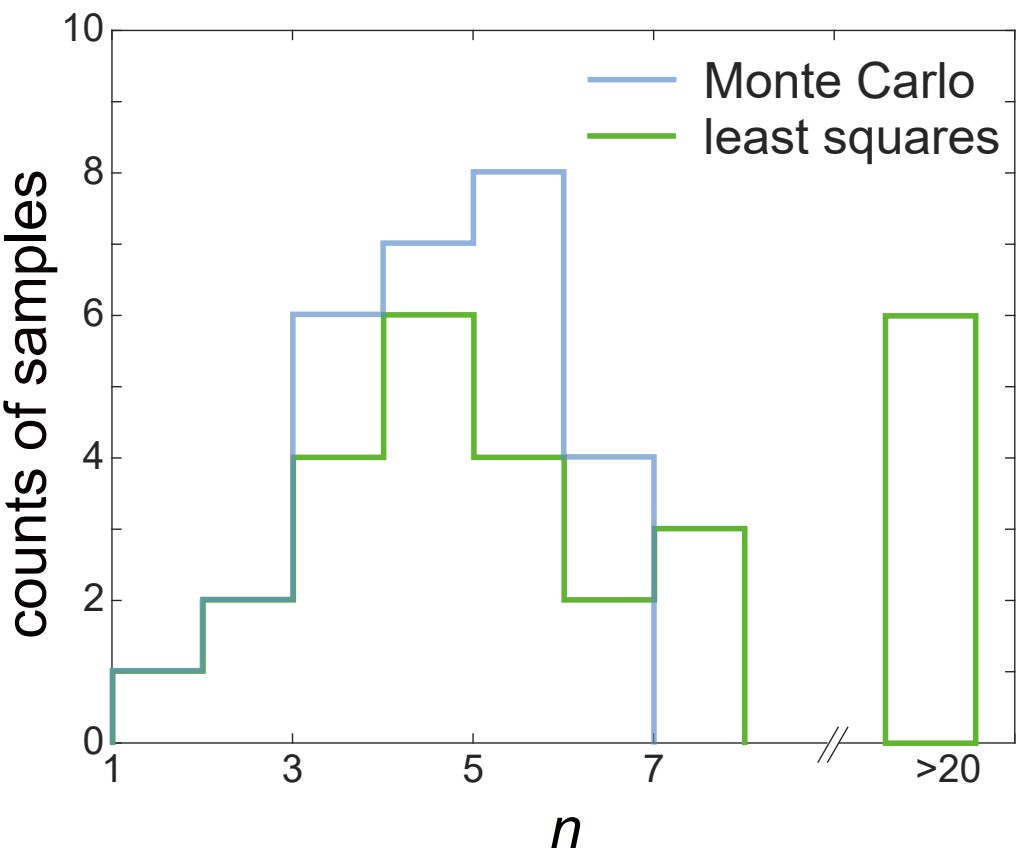

**Figure A8.** Comparison of distributions of $n$ between Monte Carlo fitting (blue line) and least squares fitting (green line). The x-axis represents fitted values of $n$ and the y-axis represents number of samples. Note that the x-axis is not continuous, jumping to greater than 20 after position of $n = 8$.

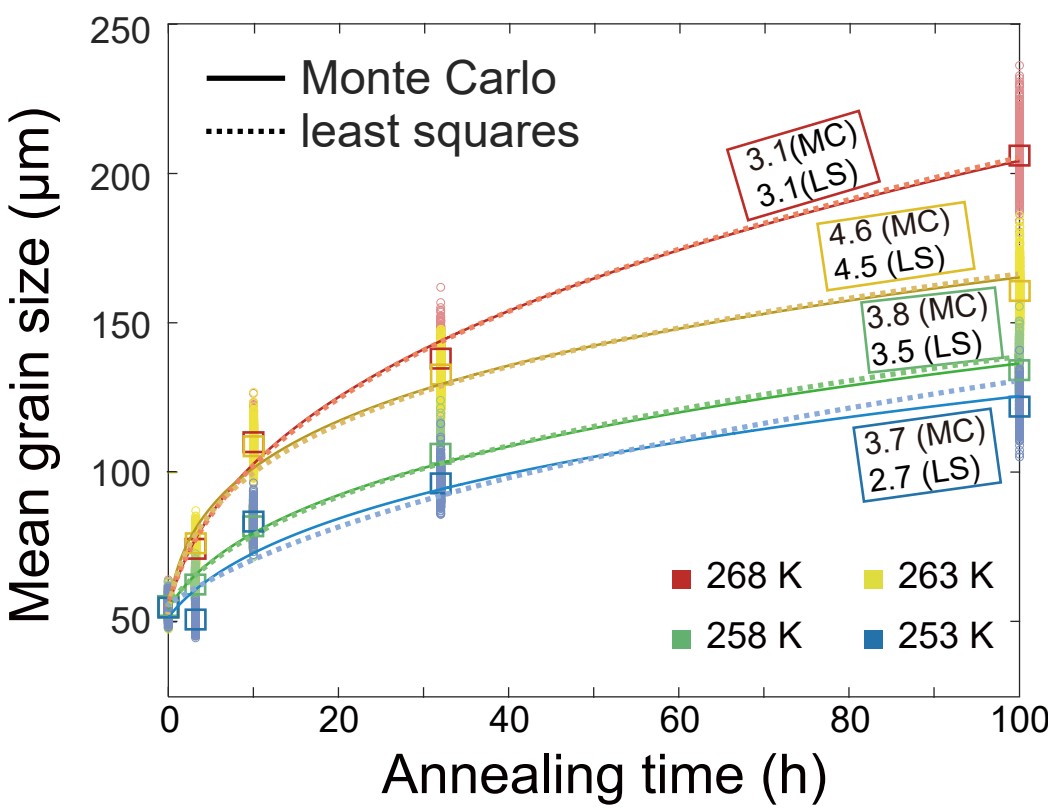

**Figure A9.** A comparison of fitting curves obtained by Monte Carlo method (colored solid line) and the least squares method (colored dashed line), using pure water ice samples as an example. The fitted values of n are presented next to each curve, with results from Monte Carlo method marked by MC and least squares method marked by LS. Different colors represent different annealing temperatures. The squares are the measured average grain size and the small circles are the grain size distribution used for fitting by the Monte Carlo method.

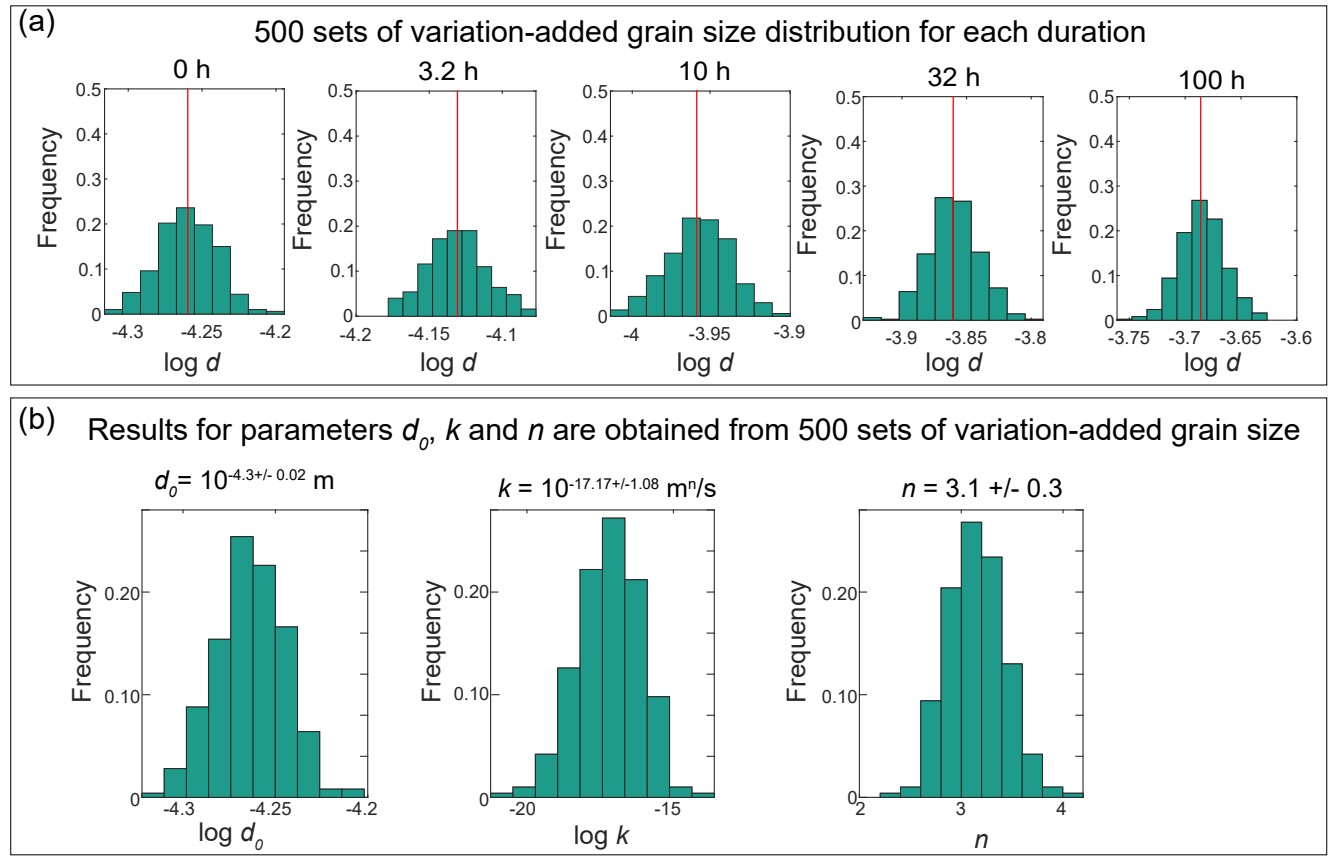

**Figure A10.** An example illustrating the Monte Carlo method. Samples of pure ice annealed at 268 K are used as examples. 500 sets of grain size data with variation were used for fitting, and corresponding fitting results were obtained. (a) For each duration, 500 sets of variation-added grain size data were generated by applying a range of random noise to the measured average grain size. In each histogram, the red vertical line represents the measured average grain size. (b) Results for parameters $d_0$, $k$ and $n$ are obtained from 500 sets of variation-added grain size. From left to right, three histograms of the parameters, $d_0$, $k$ and $n$, were illustrated. The title of each histogram presents the mean value of the corresponding parameter.

**Table A1.** Evaluation of the goodness of fits for fixed $n$.

| Temperature (K) | 268 | | | | | | | 263 | | | | | | |
|---|---|---|---|---|---|---|---|---|---|---|---|---|---|---|
| Composition | KCl | | | | MgSO$_4$ | | pure ice | KCl | | | | MgSO$_4$ | | pure ice |
| Concentration (mol/L) | $10^{-2}$ | $10^{-3}$ | $10^{-4}$ | $10^{-5}$ | $10^{-2}$ | $10^{-5}$ | | $10^{-2}$ | $10^{-3}$ | $10^{-4}$ | $10^{-5}$ | $10^{-2}$ | $10^{-5}$ | |
| fit with $n=2$   $R^2$ | 0.9 | 0.78 | 0.57 | 0.77 | 0.91 | 0.93 | 0.92 | 0.83 | 0.95 | 0.77 | 0.72 | 0.86 | 0.75 | 0.84 |
| $k$ (m$^2$/s) | $2.4\times10^{-13}$ | $2.3\times10^{-13}$ | $3.5\times10^{-13}$ | $4.3\times10^{-13}$ | $1.8\times10^{-13}$ | $1.1\times10^{-13}$ | $1.4\times10^{-13}$ | $8.7\times10^{-14}$ | $1.1\times10^{-13}$ | $8.3\times10^{-14}$ | $5.2\times10^{-14}$ | $3.4\times10^{-14}$ | $4.2\times10^{-14}$ | $9.6\times10^{-14}$ |
| fit with $n=3$   $R^2$ | 0.93 | 0.86 | 0.7 | 0.82 | 0.92 | 0.91 | 0.97 | 0.83 | 0.96 | 0.86 | 0.82 | 0.89 | 0.78 | 0.89 |
| $k$ (m$^3$/s) | $5.0\times10^{-17}$ | $5.0\times10^{-17}$ | $7.7\times10^{-17}$ | $9.8\times10^{-17}$ | $3.0\times10^{-17}$ | $1.6\times10^{-17}$ | $2.4\times10^{-17}$ | $1.5\times10^{-17}$ | $2.2\times10^{-17}$ | $1.5\times10^{-17}$ | $8.7\times10^{-18}$ | $4.5\times10^{-18}$ | $7.0\times10^{-187}$ | $1.8\times10^{-17}$ |
| fit with $n=4$   $R^2$ | 0.99 | 0.99 | 0.98 | 0.99 | 0.96 | 0.95 | 0.97 | 0.97 | 0.98 | 0.97 | 0.92 | 0.94 | 0.89 | 0.97 |
| $k$ (m$^4$/s) | $8.0\times10^{-21}$ | $8.4\times10^{-21}$ | $1.4\times10^{-20}$ | $1.9\times10^{-20}$ | $4.1\times10^{-21}$ | $1.8\times10^{-21}$ | $3.4\times10^{-21}$ | $1.9\times10^{-21}$ | $3.1\times10^{-21}$ | $1.9\times10^{-21}$ | $1.2\times10^{-21}$ | $5.0\times10^{-21}$ | $9.3\times10^{-22}$ | $2.5\times10^{-21}$ |
| fit with $n=5$   $R^2$ | 0.98 | 0.99 | 0.99 | 0.97 | 0.94 | 0.90 | 0.94 | 0.97 | 0.99 | 0.99 | 0.95 | 0.96 | 0.95 | 0.98 |
| $k$ (m$^5$/s) | $1.2\times10^{-24}$ | $1.3\times10^{-24}$ | $2.5\times10^{-24}$ | $3.6\times10^{-24}$ | $5.3\times10^{-25}$ | $2.0\times10^{-25}$ | $4.4\times10^{-25}$ | $2.2\times10^{-25}$ | $3.9\times10^{-25}$ | $2.2\times10^{-25}$ | $1.3\times10^{-25}$ | $4.6\times10^{-26}$ | $1.0\times10^{-26}$ | $3.3\times10^{-25}$ |
| fit with $n=6$   $R^2$ | 0.97 | 0.97 | 0.99 | 0.95 | 0.91 | 0.85 | 0.90 | 0.95 | 0.98 | 0.99 | 0.96 | 0.96 | 0.97 | 0.96 |
| $k$ (m$^6$/s) | $1.9\times10^{-28}$ | $2.0\times10^{-28}$ | $4.3\times10^{-28}$ | $6.7\times10^{-28}$ | $6.8\times10^{-29}$ | $2.1\times10^{-29}$ | $5.5\times10^{-29}$ | $2.5\times10^{-29}$ | $4.7\times10^{-29}$ | $2.4\times10^{-29}$ | $1.3\times10^{-29}$ | $3.9\times10^{-30}$ | $9.9\times10^{-30}$ | $3.9\times10^{-29}$ |

| Temperature (K) | 258 | | | | | | | 253 | | | | | | |
|---|---|---|---|---|---|---|---|---|---|---|---|---|---|---|
| Composition | KCl | | | | MgSO$_4$ | | pure ice | KCl | | | | MgSO$_4$ | | pure ice |
| Concentration (mol/L) | $10^{-2}$ | $10^{-3}$ | $10^{-4}$ | $10^{-5}$ | $10^{-2}$ | $10^{-5}$ | | $10^{-2}$ | $10^{-3}$ | $10^{-4}$ | $10^{-5}$ | $10^{-2}$ | $10^{-5}$ | |
| fit with $n=2$   $R^2$ | 0.9 | 0.78 | 0.57 | 0.77 | 0.91 | 0.93 | 0.92 | 0.83 | 0.95 | 0.77 | 0.72 | 0.86 | 0.75 | 0.84 |
| $k$ (m$^2$/s) | $2.7\times10^{-14}$ | $4.2\times10^{-14}$ | $4.0\times10^{-14}$ | $3.0\times10^{-14}$ | $1.3\times10^{-14}$ | $1.1\times10^{-14}$ | $5.2\times10^{-14}$ | $1.5\times10^{-14}$ | $1.6\times10^{-14}$ | $3.8\times10^{-14}$ | $4.4\times10^{-14}$ | $9.8\times10^{-15}$ | $1.1\times10^{-14}$ | $4.3\times10^{-14}$ |
| fit with $n=3$   $R^2$ | 0.93 | 0.86 | 0.7 | 0.82 | 0.92 | 0.91 | 0.97 | 0.83 | 0.96 | 0.86 | 0.82 | 0.89 | 0.78 | 0.89 |
| $k$ (m$^3$/s) | $3.3\times10^{-18}$ | $6.6\times10^{-18}$ | $7.2\times10^{-18}$ | $4.2\times10^{-18}$ | $1.2\times10^{-18}$ | $9.6\times10^{-19}$ | $7.7\times10^{-18}$ | $1.3\times10^{-18}$ | $1.6\times10^{-18}$ | $5.5\times10^{-18}$ | $7.4\times10^{-18}$ | $8.6\times10^{-19}$ | $1.1\times10^{-18}$ | $5.9\times10^{-18}$ |
| fit with $n=4$   $R^2$ | 0.95 | 0.93 | 0.83 | 0.87 | 0.93 | 0.87 | 0.98 | 0.82 | 0.97 | 0.93 | 0.91 | 0.91 | 0.78 | 0.89 |
| $k$ (m$^4$/s) | $3.5\times10^{-22}$ | $7.9\times10^{-22}$ | $9.5\times10^{-22}$ | $5.2\times10^{-22}$ | $9.8\times10^{-23}$ | $7.1\times10^{-23}$ | $9.1\times10^{-22}$ | $9.5\times10^{-23}$ | $1.4\times10^{-22}$ | $6.3\times10^{-22}$ | $9.7\times10^{-22}$ | $6.4\times10^{-23}$ | $1.1\times10^{-22}$ | $6.4\times10^{-22}$ |
| fit with $n=5$   $R^2$ | 0.96 | 0.97 | 0.91 | 0.92 | 0.92 | 0.83 | 0.96 | 0.79 | 0.96 | 0.97 | 0.96 | 0.91 | 0.82 | 0.88 |
| $k$ (m$^5$/s) | $3.2\times10^{-26}$ | $8.2\times10^{-26}$ | $9.6\times10^{-26}$ | $5.5\times10^{-26}$ | $7.2\times10^{-27}$ | $4.6\times10^{-27}$ | $9.5\text{E}10^{-26}$ | $6.0\times10^{-27}$ | $1.2\times10^{-26}$ | $5.9\times10^{-26}$ | $1.0\times10^{-25}$ | $4.1\times10^{-27}$ | $9.5\text{E}10^{-27}$ | $6.2\times10^{-26}$ |
| fit with $n=6$   $R^2$ | 0.96 | 0.96 | 0.95 | 0.94 | 0.73 | 0.34 | 0.93 | 0.52 | 0.92 | 0.97 | 0.98 | 0.23 | 0.78 | 0.85 |
| $k$ (m$^6$/s) | $2.6\times10^{-30}$ | $7.9\times10^{-30}$ | $9.2\times10^{-30}$ | $5.3\times10^{-30}$ | $4.7\text{E}10^{-31}$ | $2.9\times10^{-31}$ | $9.1\times10^{-30}$ | $3.7\times10^{-31}$ | $8.8\times10^{-31}$ | $5.3\times10^{-30}$ | $1.0\times10^{-29}$ | $7.9\times10^{-31}$ | $7.9\times10^{-31}$ | $5.5\times10^{-30}$ |