# Peer review of "Grain growth of ice doped with soluble impurities"

_EGUsphere, 2023_

## Referee Comment (RC1)

**Overview**

This study investigates the effects of soluble impurities, specifically KCl and $MgSO_4$, on grain growth in polycrystalline ice across a range of temperatures. The effects of soluble impurities on the microstructural properties of ice is crucial for understanding the dynamics of natural ice masses but is poorly understood, so this is an important and timely study. A particularly interesting finding is that temperature relative to the eutectic has a more significant effect on grain growth than the actual concentration of impurities, which seems to have little impact.

The manuscript is clearly written, with a thorough discussion. Some fairly minor points should be addressed, as outlined below.

I enjoyed reading this manuscript, and I hope these comments are useful.

**General comments**

Generally, I think it's important to acknowledge in the manuscript that these results concern a specific scenario (relatively high concentrations of two soluble impurities), and can't be easily generalised to describe the behaviour of all natural ice. So I would stop short of recommending an 'established impurity factor', as in lines 16 and 483. The results are still very valuable because they expose the relationships between grain size, temperature and concentration of these specific impurities.

In a similar vein, it would be good to see some justification of why these particular ionic species and concentrations were chosen for the study. The concentrations are high compared to what is usually seen in ice sheets (Cuffey and Paterson [2010] give a range of 1e-7 to 1e-5 mol/L total dissolved impurity concentration in most polar ice sheets, with higher values closer to the bed). While overall it seems like concentration is not the most important factor, there must be a lower limit where there is no longer an effect on grain growth? There should be some more discussion of how these results might or might not scale to natural ice masses.

The same goes for the choice of KCl and $MgSO_4$. It's clear that different ions have different effects on grain size, so how might the results of this study translate to describing ice with both of these together, along with other impurities as well?

In the Results, some grain size distributions are described as having a 'bimodal component' (lines 172, 185). It's not obvious to me exactly what this means. E.g., the 10-100hr annealed KCl sample in Figure 2(b) does have some higher frequencies of larger grains, but this second 'mode' isn't any larger than one seen in the smaller grain sizes of the starting material in Figure 2(d), which is described as normally distributed. The starting materials in general look to me to have a more skewed distribution, compared to the annealed samples which are more normally distributed. I don't think this is important to the conclusions, but I would recommend either quantifying the skewness and bimodality of the distributions, or removing these descriptions.

**Specific comments**

**line 75:** 'Overestimation of the value of $n$ can occur if grain size distribution not in steady state...': are the grain size distributions in this study in steady-state? Could that bias the $n$ value estimates?

**line 6:** For glaciologists with poor mental maths (like me), it would be good to include what these °K values correspond to in °C.

**line 25:** 'Demonstrate weaker mechanical properties' seems like a roundabout way of saying 'are mechanically weaker'

**line 33:** 'concentration' is probably more accurate than 'content'

**line 76:** 'transition in grain boundary structure': it's not clear what this means exactly

**line 83:** What is the 'prepared solution', is it the solution from the previous sentence?

**line 100:** Briefly mention which time periods were chosen up to 100 hrs (i.e. 3.2hrs, 10 hrs, 32 hrs) and why (to capture rapid change in grain size early in the annealing process?)

**line 147-149:** The starting material grain size distribution in (c) of each of those figures looks shorter and broader than the others. Are they all cut from the same starting sample? I doubt it would make much difference, but it's worth making a brief comment on how much variation there is between the starting material samples.

**line 178:** 'Given an annealing time...' does this refer to any length of time, or a specific one?

**line 195:** 'the growth of grains displays self-similarity'; does this just mean they are similar to one another?

**line 223:** Isn't this contradictory? Or there is a difference in grain size between different concentrations, but it's not statistically significant?

**Figure 8:** This is a great visualisation. The square symbols are a little hard to interpret, it would help to have the line width the same between all of them (rather than scale the line width with the size of the square). It might even be better to have a separate panel for the $MgSO_4$ values, so that they can be viewed side-by-side.

**lines 232-233:** 'could be deemed as'; I think you can just say that it is relatively independent, and therefore you have decided to treat the average value as representative of all of the samples going forward.

**line 384:** Hammonds and Baker [2016] also found the same with $Ca^+$-doped ice... maybe worth mentioning.

**line 472:** There was one case with $n = 1.1$ ($10^{-5}$mol/L $MgSO_4$ annealed at 258K.) Was this discarded as an outlier?

**Appendix B:** It would be useful to have the full grain size datasets also included in the figshare repository, so that others don't have to repeat the Cellpose and ImageJ analysis to use the data.

**Typos, etc.**

**line 25:** In **the** grain-size sensitive...

**line 41:** Replace the semicolon with a full stop

**line 47:** Highly concentrated NaCl solutions

**line 53:** 'including' makes it sound as though there were other impurities as well. I'd go with 'soluble impurities (potassium chloride and magnesium sulfate)...'

**line 94:** ice particles **were** pressed

**line 104:** **was** done

**line 110:** within **a** few minutes

**line 121:** missing full stop

**line 129:** and **the** standard deviation?

**line 143:** remove 'is'

**line 167:** microstructures

**line 168:** temperature

**line 183:** 'for' rather than 'to' 100 hours

**line 187:** 'increase' is repeated

**line 212:** microstructures

**line 229:** **is** illustrated

**line 256:** remove 'obviously'

**line 391:** separately

**line 428:** impurities

**line 481:** soluble **impurities**

**line 525:** illustrated

**line 539:** to directly correlate

**References**

K. M. Cuffey and W. S. B. Paterson. *The physics of glaciers*. Butterworth-Heinemann, Burlington, MA, 2010. ISBN 978-0-12-369461-4. doi: 10.1016/0016-7185(71)90086-8. arXiv: 1011.1669v3 ISSN: 00167185.

K. Hammonds and I. Baker. The effects of Ca++ on the strength of polycrystalline ice. *Journal of Glaciology*, 62(235):954–962, 2016. ISSN 00221430. doi: 10.1017/jog.2016.84. URL `http://www.journals.cambridge.org/abstract_S0022143016000848`.

---

## Author Response (AR1)

Dear Dr. McCormack,

Thank you for your time and effort! We appreciate the helpful comments from two reviewers and you. Here we present our responses to the comments. Our responses are in black; comments are in blue. Revisions of the manuscript are in italics. Changes that are different from our posted responses are in orange.

**Reviewer 1 Dr. Craw**

**Response to general comments**

Generally, I think it's important to acknowledge in the manuscript that these results concern a specific scenario (relatively high concentrations of two soluble impurities), and can't be easily generalised to describe the behaviour of all natural ice. So I would stop short of recommending an 'established impurity factor', as in lines 16 and 483. The results are still very valuable because they expose the relationships between grain size, temperature and concentration of these specific impurities.

Reply: Thank you very much for your suggestion, we will remove these statements in the abstract and conclusion sections. We will also add statements that underscore the limit of our experiments, i.e., high concentrations of two soluble impurities, that cannot accurately mirror natural conditions, i.e., low concentrations of a diverse range of soluble and insoluble impurities.

In a similar vein, it would be good to see some justification of why these particular ionic species and concentrations were chosen for the study. The concentrations are high compared to what is usually seen in ice sheets (Cuffey and Paterson [2010] give a range of $1e^{-7}$ to $1e^{-5}$ mol/L total dissolved impurity concentration in most polar ice sheets, with higher values closer to the bed). While overall it seems like concentration is not the most important factor, there must be a lower limit where there is no longer an effect on grain growth? There should be some more discussion of how these results might or might not scale to natural ice masses.

The same goes for the choice of KCl and MgSO4. It's clear that different ions have different effects on grain size, so how might the results of this study translate to describing ice with both of these together, along with other impurities as well?

Reply: In response to your first question concerning the presence of a lower limit for the impact of impurity concentration on grain growth, we suggest that it can be evaluated in two distinct scenarios: those above the eutectic point and those below the eutectic point.

(1) **Above the eutectic point**. In scenarios above the eutectic point, our dataset is relatively limited, because this study only includes data related to KCl-doped ice annealed at 268 K (-5°C). Results from our experiments indicate that, even at the minimum concentration of KCl utilized in this study, specifically at $1\times10^{-5}$ mol/L, there remains a notable promotion of grain growth. It is crucial to acknowledge that these minimum concentrations utilized in our experiments are still higher than ice sheets, specifically at $1\times10^{-7}$ to $1\times10^{-5}$ mol/L (Cuffey and Paterson, 2010). Hence, in the revised manuscript, we will add statements to emphasize that results from our experiments do not fully represent natural scenarios; caution is needed when extrapolating our experimental results to warmer glacial regions.

The bed of glacier is characterized by relatively high temperatures and impurity content (close to melting point in some ice cores and total impurity concentration of $\sim 8 \times 10^{-4}$ mol/L, according to Cuffey and Paterson (2010)). Thus, the potential enhancement of grain growth due to chemical impurities could be considered. However, the height range of the glacier bed (Cuffey, 2000) is limited and is not the focus of glaciologists.

(2) **Below the eutectic point**. In this scenario, our experiments encompass two sets of KCl-doped ice, each maintained at least 2 K below the eutectic point, and three sets of $MgSO_4$-doped ice maintained at least 5 K below the eutectic point.

For KCl, despite the rejection of ions by ice crystals, previous experimental observations and simulations have indicated that $Cl^-$ and $K^+$ can be incorporated into the crystal lattice with low concentrations under conditions below the eutectic point. It is only when the concentration surpasses the solubility threshold (approximately 2 to 3 $\times 10^{-5}$ mol/L, according to Yashima et al., (2021)) that hydrates precipitate at the grain boundaries, thereby impeding grain growth. This solubility value may be above the average content range of Earth's glaciers ($1 \times 10^{-7}$ - $1 \times 10^{-5}$ mol/L), suggesting that the impact of chloride salts on grain growth might be small in cold glacial regions.

In this study, ice doped with $MgSO_4$ has a lowest concentration of $1 \times 10^{-5}$ mol/L. At this concentration the grain growth was still impeded. The solubility of $MgSO_4$ in ice may be lower than this value. Currently, there is limited research on $MgSO_4$ solubility in ice. We assume it approximates the actual solubility, which may be lower than $2 \times 10^{-6}$ mol/L (Iliescu and Baker, 2008). Additionally, the eutectic point of sulfate is generally high, placing glaciers below this temperature. In ice deposited during Pleistocene glacial climates with higher impurity concentrations ($10^{-6}$ to $10^{-5}$ mol/l, according to Cuffey and Paterson, 2010), sulfate may moderately hinder grain growth. This conclusion appears to establish a connection between reduced ice core grain size and elevated concentrations of soluble impurities (sulfate and chloride ions) (e.g., Alley 1986, Langway 1988, Paterson 1991).

Next, we explore how the results of this study can be applied to describing ice containing other impurities. In this research, KCl was chosen to exemplify common alkali metal chloride, while $MgSO_4$ was selected to represent prevalent alkali metal sulfates, found in glaciers (Cuffey and Paterson, 2010). As mentioned in the Introduction, above the eutectic point, soluble impurities manifest as saturated solutions at grain boundaries, facilitating grain boundary migration. Consequently, when extrapolating our outcomes to the impact of other salts on ice grain growth, their respective eutectic points become a crucial consideration. In conditions below the eutectic point, soluble impurities exceeding the solubility range reside at grain boundaries in the form of hydrates, impeding grain boundary migration. Therefore, when extending our findings to ice grain growth influenced by other salts, reliance on the solubility of the salt in the ice lattice may be pertinent.

We will add the above discussion in Section 5.5: Implications for natural ice. (Compared to our posted response in the interactive discussion system, two transitional sentences were added to the beginning of this paragraph.)

"*Based on the experimental findings, soluble impurities exhibit a discernible influence on grain growth. Nonetheless, given the substantial disparities in temperature, impurity types, and*

*concentrations between our experimental conditions and those of natural glaciers, a more meticulous discussion is warranted when extrapolating our results to natural glacier settings. $Cl^-$ and $SO_4^{2-}$ constitute two of the most prevalent anions in polar glaciers, while $Na^+$, $Mg^{2+}$, $Ca^{2+}$, and $K^+$ are the predominant alkali metal cations in Earth's glaciers (Obbard and Baker, 2007). In this investigation, we have chosen KCl and $MgSO_4$ as representatives of chloride salts (hereinafter specifically referred to as chloride salts of the above four ions) and sulfates (hereinafter specifically referred to as sulfates of the above four ions), respectively. The impacts of KCl and $MgSO_4$ on ice grain growth can be attributed to their eutectic points and solubility within the crystal lattice of ice. In extrapolating our results to the influence of chloride or sulfate on ice grain growth, it is imperative to consider these two aspects. We suggest that it can be evaluated in two scenarios: above the eutectic point and below the eutectic point.*

*(1) In scenarios above the eutectic point, our dataset is relatively limited, because this study only includes data related to KCl-doped ice annealed at 268 K (-5°C). Our results reveal a significant promotion of grain growth by soluble impurities at temperatures above the eutectic point. This phenomenon is also observed in NaCl-doped ice (De Achaval et al. 1987). Except for KCl, other chloride salts exhibit relatively low eutectic points (approximately 223 K to 252 K, according to Baumgartner (2009)), potentially nearing or even falling below the temperature of some glaciers (Cuffey et al.,2000). In other words, most glaciers have temperatures above the eutectic point of chloride salts. Nevertheless, the lowest concentration utilized in our experiments ($1\times10^{-5}$ mol/L) remains higher than the total impurity concentration in the ice sheet, estimated at about $1\times10^{-7}$ to $1\times10^{-5}$ mol/L (Cuffey and Paterson, 2010, pp.33-34). Applying our experimental results directly to glaciers with low impurity concentrations poses challenges. Consequently, extra caution should be paid when extrapolating our results to lower impurity concentrations in glaciers.*

*(2) At temperatures below the eutectic point, we hypothesize that salt hydrates will precipitate at grain boundaries, impeding grain growth only when the concentration surpasses the solubility in the crystal lattice of ice. Given that $Cl^-$ ions can integrate into the interior of the ice crystal lattice, chloride salts exhibit slightly greater solubility in ice than sulfate salts. Specifically, the solubility of KCl is approximately 2 to $3\times10^{-5}$ mol/L (Yashima et al., 2021), while NaCl's solubility ranges from $3\times10^{-5}$ to $1\times10^{-4}$ mol/L (Mongnant et al., 2001, and Gross, 1977). For $Ca^{2+}$ and $Mg^{2+}$ with smaller ionic radii, we assume that the solubility of their corresponding chlorine salts ($CaCl_2$ and $MgCl_2$) in the ice lattice is of the same order as KCl and NaCl, that is, at least greater than $1\times10^{-5}$ mol/L. Thus, the solubility of common chloride salts may generally exceed the average content range of glaciers ($1\times10^{-7}$ to $1\times10^{-5}$ mol/L). Therefore, the effect of chloride salts on grain growth in colder glaciers is minimal. In contrast to KCl, grain growth remains impeded in the ice sample doped with $MgSO_4$ even at the lowest concentration ($1\times10^{-5}$ mol/L). We assume the solubilities of sulfates are similar to the solubility of sulfate ion, which may be lower than $2\times10^{-6}$ mol/L (Iliescu and Baker, 2008). Additionally, the eutectic points of sulfates are generally high (>268 K), placing glaciers below these temperatures. In ice deposited during Pleistocene glacial climates with higher impurity concentrations ($10^{-6}$ to $10^{-5}$ mol/L, according to Cuffey and Paterson, 2010), sulfates may hinder grain growth. This conclusion could establish a connection between reduced grain size in ice cores and elevated concentrations*

*of soluble impurities (sulfate and chloride ions) (e.g., Alley 1986, Langway 1988, Paterson 1991)."*

In the Results, some grain size distributions are described as having a 'bimodal component' (lines 172, 185). It's not obvious to me exactly what this means. E.g., the 10-100hr annealed KCl sample in Figure 2(b) does have some higher frequencies of larger grains, but this second 'mode' isn't any larger than one seen in the smaller grain sizes of the starting material in Figure 2(d), which is described as normally distributed. The starting materials in general look to me to have a more skewed distribution, compared to the annealed samples which are more normally distributed. I don't think this is important to the conclusions, but I would recommend either quantifying the skewness and bimodality of the distributions, or removing these descriptions.

Reply: Upon a thorough review of our results, we found that the presence of a bimodal distribution is not apparent. Furthermore, we omitted any discussion of bimodal distributions in the subsequent sections. So, we have chosen to exclude these descriptions from our analysis. Now the sentences are changed to the following:

Line 157: "*The grain-size distribution remains log-normal, and in the sample annealed for 100 hours, there is higher frequencies at larger grain sizes (500 to 1000 µm).*"

Line 173: "*while a few samples (such as the sample with $10^{-5}$ mol/L KCl annealed for 10 to 100 hours, Figure 2(b)) exhibit higher frequencies at grain sizes larger than the peak.*"

Line 186: "*The grain size distribution remains lognormal after annealing. In the sample with $10^{-5}$ mol/L KCl annealed to 100 hours, the peak frequency is slightly lower and the tail towards larger grain sizes is longer compared to other samples.*" (Compared to our posted response in the interactive discussion system, this sentence was modified for easier reading.)

Line 177: We decided to delete the following sentence.

"The increase in $\sigma_d/d$ usually suggests some grains grow faster than others, resulting a bimodal distribution in grain size."

**Response to Specific comments**

line 75: 'Overestimation of the value of n can occur if grain size distribution not in steady state...': are the grain size distributions in this study in steady-state? Could that bias the n value estimates?

Reply: The statement came from the citation Breithaupt et al., (2021). They assess whether the distribution of grain size remains steady. The grain size distribution at the steady state follows a log-normal pattern. After normalization, the grain size distributions at various annealing times exhibit similarity, that is, they have the same or similar normalized variance. The steady-state distribution appears independent of the initial grain size distribution and eventually converges to a distinctive normalized pattern. As such, we consider that our grain size distribution is in a steady state.

Not all of our grain-size distributions display characteristics of a steady-state distribution. However, upon a thorough review of Breithaupt et al., (2021), I did not find explicit information linking the steady-state distribution to the value of *n*. Therefore, we removed the description of the relationship between the steady-state distribution and the value of *n*.

line 6: For glaciologists with poor mental maths (like me), it would be good to include what these K values correspond to in °C.

Reply: We have modified the descriptions in the Section Abstract and Methods (lines 6 and lines 100).

Line 6: "*Samples were annealed for a maximum of 100 h at a hydrostatic confining pressure of 20 MPa (corresponding to a depth of about 2 km) and different constant temperatures of 268, 263, 258 and 253 K (corresponding to -5, -10, -15 and -20°C, respectively).*"

Line 100: "*Annealing was conducted at a hydrostatic pressure of 20 MPa, corresponding to the pressure at the base of a 2-km glacier, and temperatures of 268, 263, 258 and 253 K (corresponding to -5, -10, -15 and -20°C, respectively), to a maximum duration of 100 h.*"

line 25: 'Demonstrate weaker mechanical properties' seems like a roundabout way of saying 'are mechanically weaker'.

Reply: we have revised the statement accordingly. (lines 25)

"*In the grain-size sensitive deformation regime, ice samples with smaller grain sizes are mechanically weaker, resulting in reduced stresses under constant strain rates or increased strain rates under constant stresses (Goldsby and Kohlstedt, 2001).*"

line 33: 'concentration' is probably more accurate than 'content'.

Reply: We have replaced 'content' with 'concentration' in the text (lines 33).

"*However, some other studies reported that a high concentration of soluble impurities inferred from electrical conductivity measurements does not necessarily result in a slowdown of grain growth (Durand et al., 2006).*"

line 76: 'transition in grain boundary structure': it's not clear what this means exactly.

Reply: In the review by Cantwell et al., (2014), facet coarsening, changes in orientation, changes in grain boundary thickness, and amorphous phase grain boundaries are all classified as complexion transitions in grain boundaries. These transformations significantly impact the rate of grain boundary migration. However, due to the absence of TEM evidence in our experiments, an accurate assessment of our grain boundary structure is challenging. Moreover, as this information is not clear linked to the subsequent results and discussions, and considering its potential to mislead readers, we have decided to delete the relevant content.

line 83: What is the 'prepared solution', is it the solution from the previous sentence?

Reply: We have revised the statement (lines 83).

"*Second, the prepared KCl (or MgSO₄) solution was added to the container of a medical ultrasonic nebulizer, which generated a fine mist of droplets with a diameter less than 50 μm.*"

line 100: Briefly mention which time periods were chosen up to 100 hrs (i.e. 3.2hrs, 10 hrs, 32 hrs) and why (to capture rapid change in grain size early in the annealing process?)

Reply: We have modified the descriptions in the Section Methods (lines 100).

"*Furthermore, annealing experiments were conducted with durations of 3.2 hours, 10 hours, and 32 hours to capture the rapid changes in microstructure and grain size in the early stage of annealing.*"

line 147-149: The starting material grain size distribution in (c) of each of those figures looks shorter and broader than the others. Are they all cut from the same starting sample? I doubt it would make much difference, but it's worth making a brief comment on how much variation there is between the starting material samples.

Reply: We did not conduct repeated observations on the 0-h sample, i.e., annealing experiments at different temperatures use the same 0-h samples. In panel (c), the 0-h sample corresponds to the one doped with $10^{-2}$ mol/L MgSO₄. The higher proportion of small-sized grains in this sample gives its initial grain size distribution a broader appearance compared to the distributions of the other samples 0-h samples. The larger molecular mass of MgSO₄ (120), compared to KCl (molecular mass 74.5), contribute to this difference. Vetráková et al., (2019) observed that higher-concentration solution result in smaller average ice particle sizes when frozen in liquid nitrogen. Therefore, we speculate that the ice particles formed in liquid nitrogen by doping a $10^{-2}$ mol/L MgSO₄ solution are generally smaller, leading to a broader and wider particle size distribution towards the finer sizes.

As described in our results, the grain size distribution of samples which doped with $10^{-2}$ mol/L MgSO₄ (corresponding to Figure 2 (c) to Figure 5 (c)) follow log-normal distribution. And after normalization, the grain size distributions at different annealing times exhibit similarities, indicating a steady-state distribution. The steady-state distribution appears to be independent of the initial grain size distribution and eventually converges to a distinct normalized pattern. Consequently, we consider that the initial grain-size distribution does not have a significant impact on the grain size distribution of the annealed sample.

line 178: 'Given an annealing time...' does this refer to any length of time, or a specific one?

Reply: We have modified the descriptions in the text. (After the comments from Dr. McCormack, the sentences in lines 178, 206 and 223 are modified from the version posted in the interactive discussion.)

Line 178: "*At any given time during annealing, grain sizes in all KCl-doped ice are coarser than those in pure ice, as illustrated in Figure 6(a).*"

Line 189: "*At any given time during the 100-h annealing, grain sizes of ice samples doped with different concentrations of KCl are similar to each other and also similar to the grainsize in pure ice, as illustrated in Figure 6(b).*"

Line 196: "*At any given time, grain sizes in all KCl-doped ice are slightly finer than those in pure ice, as illustrated in Figure 6(c).*"

Line 206: "*At any given time, grain sizes in ice doped with higher concentrations ($10^{-2}$ and $10^{-3}$ mol/L) are significantly finer than those in ice doped with lower concentrations ($10^{-4}$ and $10^{-5}$ mol/L), as illustrated in Figure 6(c).*"

Line 223: "*At any given time during the annealing, grain sizes in $MgSO_4$-doped ice are always finer than those in pure ice, as illustrated in Figure 7(b), (c) & (d).*"

line 195: 'the growth of grains displays self-similarity'; does this just mean they are similar to one another?

Reply: Based on Faul and Scott (2006), 'self-similarity' entails 'grain size distribution normalized by the mean grain size is time invariant'. We apologize for the misleading writing; we have revised related description in the text.

Lines 130: "*The normalized standard deviation ($\sigma_d/d$) can be utilized to evaluate the self-similarity of the grain size distribution, i.e., the grain size distributions are similar to each other (Faul and Scott, 2006).*"

lines 195: "*The grain size distributions among the samples are similar to each other, suggested by the values of $\sigma_d/d$, keeping in the range between 0.4 and 0.6 for all samples.*"

Lines 204: "*The growth of grains are similar to each other, as the values of ($\sigma_d/d$) vary within the range between 0.4 and 0.6 for all samples.*"

Lines 221: "*The growth of grains are similar to each other, as the values of ($\sigma_d/d$) vary within the range between 0.4 and 0.55 for all samples.*"

line 223: Isn't this contradictory? Or there is a difference in grain size between different concentrations, but it's not statistically significant?

Reply: To prevent any potential confusion among readers, we have revised the sentence (lines 223).

"*At 263 K and 253 K, samples with high impurity concentration ($10^{-2}$ mol/L) exhibited slightly smaller grain sizes than the low-concentration sample ($10^{-5}$ mol/L). Overall, the diversity in grain size between different concentrations is not evident.*"

Figure 8: This is a great visualisation. The square symbols are a little hard to interpret, it would help to have the line width the same between all of them (rather than scale the line width with the size of the square). It might even be better to have a separate panel for the $MgSO_4$ values, so that they can be viewed side-by-side.

Reply: We apologize for the unclear illustration. We have updated Figure 8 to include separate value panels for KCl and $MgSO_4$, respectively. The updated Figure 8 and caption is presented below.

[Figure]

"*Figure 8. Plots of grain size exponent, n, versus temperature and sample type. Round symbols are KCl-doped ice, and square symbols are MgSO$_4$-doped ice. The symbols within the dashed rectangle represent ice doped with the same concentration and annealed at the same temperature. Symbols are colored with respect to the value of n. Larger size and warmer color represent a larger value of n.*"

lines 232-233: 'could be deemed as'; I think you can just say that it is relatively independent, and therefore you have decided to treat the average value as representative of all of the samples going forward.

Reply: We have modified the descriptions in the text (lines 232). After the comments from Dr. Gerbi, these sentences are modified to the following.

"*While no systematic changes were observed in the value of n as the solute concentration or temperature changed, we cannot rule out that the uncertainty in the value of n may be higher than the systematic changes due to solute concentration and temperature. Thus, at this stage, the average value of n can be regarded as representative of all the samples.*"

line 384: Hammonds and Baker [2016] also found the same with Ca$^{2+}$-doped ice... maybe worth mentioning.

Reply: Hammonds and Baker (2016) conducted an analysis of grain size following deformation tests with CaSO$_4$-doped ice. They observed that, in compressive deformation experiments with the lowest applied strain rate of $1\times10^{-6}\times s^{-1}$, the average grain size of CaSO$_4$-doped samples was larger than that of pure water ice. While no difference in grain size was noted between compression and tensile tests at higher deformation rates, intergranular cracks were evident in these samples. Similarly, Hammonds and Baker (2018) observed, in the deformation test with H$_2$SO$_4$-doped ice, that several groups of samples developed intergranular microcracks, with no discernible difference in grain size between doped ice and pure water ice. Notably, in samples without intergranular microcracks, the grain size of doped ice was significantly larger than that

of pure ice. As a result, we cautiously speculate that the presence of intergranular microcracks observed by Hammonds and Baker (2016) might impede the migration of grain boundaries rather than being promoted by doped impurities. Consequently, we suggest comparing Hammonds & Baker (2016) with our study may mislead readers.

line 472: There was one case with $n = 1.1$ ($10^{-5}$ mol/L MgSO4 annealed at 258 K) . Was this discarded as an outlier?

Reply: The observed value of $n = 1.1$ is lower than the theoretical value of $n = 2$, indicating an outlier in our data. We speculate that this lower $n$ value may be attributed to a slow grain growth rate, resulting in difficulties in fitting. We added a sentence in Section 5.1 to address this.

*"Fitting of data from this sample suggests n = 1.1, which is lower than the theoretical value of 2. We think this low value of n may be attributed to the slow growth in the first 10 h. Because experiments are not perfect, such variations in the fitted values of n were also reported in previous studies (Azuma et al., 2012)."*

Appendix B: It would be useful to have the full grain size datasets also included in the figshare repository, so that others don't have to repeat the Cellpose and ImageJ analysis to use the data.

Reply: Thank you for bringing this up. We have uploaded the full grain size datasets to the Figshare repository.

**Response to Typos**

Thank you very much for identifying grammatical errors and spelling issues. We have addressed each of them individually in the revised text (as indicated below).

line 25: In the grain-size sensitive...

[revised manuscript text omitted]

**Reviewer 2 Dr. Gerbi**

1. Many places in the manuscript and in the conclusions (e.g., lines 198, 209, 230, 231, 286, 294, 472, 473) n is described as varying. Yet Line 232 (similar to line 364) claims that "n can be regarded as relatively independent of both temperature and concentration". I find those statements incompatible, but perhaps I am missing something. In my limited time spent with these data, I do not see a systematic relationship, but I have difficulty seeing n as independent, particularly given how far outside of uncertainty the values lie according to Table 1. Possible resolutions are that the uncertainties are higher than reported or that the data aren't able to capture some additional complexity.

Reply: Thank you for the suggestion. We found no systematic changes in values of n with changes in solute concentration or temperature. However, the values of $n$ of some samples showed significant uncertainty, and we cannot rule out the impact of this uncertainty on the value of $n$, although our fitting method strives to minimize the uncertainty of the value of $n$. As you suggested, we don't use "independent" to describe it.

Uncertainties in $n$ are also brought by bubbles. The amount and size of air bubbles trapped along grain boundaries can strongly affect the value of $n$ (Azuma et al., 2012). In the study of Azuma et al. (2012), the value of $n$ of bubble ice varied between 6 and 20, with the majority lied between 6 and 11. With bubbles inducing such large uncertainties, we agree that a potential trend, if existed, would be overwritten by the uncertainties. But for this study, we have to work with the current data and acknowledge the weakness.

Now the sentences are changed to the following:

Line 232: "*While no systematic changes were observed in the value of n as the solute concentration or temperature changed, we cannot rule out that the uncertainty in the value of n may be higher than the systematic changes due to solute concentration and temperature.*"

We also modified the first paragraph in Section 5.4:

Line 362: "*The presence of air bubbles could induce large uncertainties in the fitting of n (Azuma et al., 2012), which may overshadow the effect of different impurities contents on n, if existed. However, in this study, we have to make inferences based on the available data set, while acknowledging the uncertainties. Given the same experimental conditions, we assume the effect of air bubbles should apply equally to pure and doped ice samples, and thus, we can investigate the effect of soluble impurities on grain growth by comparing samples with different impurity concentrations. As elucidated in the previous subsections, the values of n suggest that the grain growth occurred in a multi-phase system. We have not observed any evident systematic trends in n with changes in temperature or solute concentration. We tentatively assume that the grain growth in all doped ice samples can be considered as controlled by the same mechanism. As illustrated in Appendix Figure A8, the majority of fitted values of n lie around 4 and 5. And, as illustrated in Table A1, n = 4 provides good fits for all samples. We take n = 4 as a representative value for all samples in the following discussions.*"

We also expanded Appendix Table A1. Values of $k$ for $n = 2$ to $6$ are all presented.

2. Throughout the manuscript, including in the conclusions, n is characterized as having an average value. I do not have a sense of how that average is calculated, and how much the experimental condition choices affect the determination of that average. Meaning, if 5 experiments had been run above the eutectic but still in the same temperature range, I expect the average n would be different. So it is hard for me to see how reliable the average is.

Reply: We apologize for the misleading usage of the word "average". In fact, 4 was used as a representative value for $n$. It is inevitable to acknowledge the presence of systematic errors in many stages of the experiment, and ultimately, this will also be reflected in the values of $n$. We agree with you that the average value does not hold significant meaning across different samples. Therefore, we have decided to modify the value of $n$ to a range in the conclusion, rather than an exact average. However, we are still considering retaining $n = 4$ in our discussions, specifically for comparing the crystal growth constants (values of $k$) of samples with different compositions. As illustrated in Table A1 and newly added Figure A7, we have tested the goodness of fitting and calculated $k$ values for different values of $n$ from 2 to 6. We found that the variation in $k$ values between different samples increased with increasing values of $n$. However, with different values of $n$, the order of $k$ between different samples exhibits a consistent pattern. This also indicates that the choice of $n$ values does not significantly affect the observed pattern of grain growth rates among different samples. We find that $n = 4$ gives reliable fits for all temperatures and impurity concentrations. As illustrated in the newly added Figure A8, the majority of fitted values of $n$ lie between 3 to 6. Since the comparison for grain growth rate requires a common value of $n$, 4 is the best choice for further interpretation based on the available data set. We have revised the first paragraph of section 5.4, as described in the reply of comment 1.

We have added the following figures and captions to the Appendix.

[Figure]

**Appendix A7**: Temperature dependence of the grain-growth rate constant $k$ of KCl-doped ice (left column) and MgSO4-doped ice (right column) calculated by fixing different values of $n$. (a) $n = 2$, (b) $n = 3$, (c) $n = 5$, (d) $n = 6$. Data for pure ice were plotted as black squares in both panels as a reference. Different colors represent different impurity concentrations. The black line is the fit of pure ice.

[Figure]

**Appendix A8:** Comparison of distributions of $n$ between Monte Carlo fitting (blue line) and least squares fitting (green line). The x-axis represents fitted values of $n$ and the y-axis represents number of samples. Note that the x-axis is not continuous, jumping to greater than 20 after position of $n = 8$.

[Figure]

**Appendix A9:** A comparison of fitting curves obtained by Monte Carlo method (colored solid line) and the least squares method (colored dashed line), using pure water ice samples as an example. The fitted values of $n$ are presented next to each curve, with results from Monte Carlo method marked by MC and least squares method marked by LS. Different colors represent different annealing temperatures. The squares are the measured average grain size and the small circles are the grain size distribution used for fitting by the Monte Carlo method.

We have also revised the conclusion.

Line 471: "*The values of the best-fit grain growth exponent n lie between 2.6 to 6.2.*"

Line 473: "*The grain growth exponent obtained from pure ice samples ranges from 3.1 to 4.6.*"

3. The manuscript spends significant time discussing the grain growth constant, *k*, yet no conclusion point relates to this value.

Reply: We have modified the descriptions in the Conclusions.

Line 472: *"The grain growth exponent for pure ice samples ranged from 3.1 to 4.6. Compared to previous studies, we observed that the grain growth rate in the laboratory is lower than that of artificial pure ice; and the activation energy for grain growth is close to that of glacial ice and artificial ice containing smaller bubbles. This may be because some bubbles still exit in our samples, although they are compressed by the confining pressure to sizes below the detection limit of our optical microscope."*

Line 475: *"Above the eutectic point, grains doped with soluble impurities exhibit a higher grain growth constant, compared to pure water ice, i.e., faster grain growth rate. Below the eutectic point, ice doped with a specific concentration of soluble impurities manifests a smaller grain growth constant, compared to pure water ice, i.e., slower grain growth rate."*

4. In conclusion point #4, I suggest separating the observations from the inferred cause (i.e., that soluble impurities enhance grain growth above the eutectic and the presence of a melt phase; similar comment for below the eutectic).

Reply: Thank you for the suggestions. We have separated them in different points. The third conclusion point is the same as in our response to question #3. The inferred cause was added as a new conclusion point.

*"The enhancement in grain growth at temperatures above the eutectic point could be attributed to the formation of a molten phase by the doped salt. The inhibition in grain growth at temperatures below the eutectic point may be attributed to the formation of hydrates at grain boundaries."*

5. In Figures 6, 7, and 9, please describe how the best fits are calculated.

In the Methods Section we introduce how to perform the best fit. Here, we provide a more in-depth explanation of the Monte Carlo fitting process and comparisons with the outcomes from the least squares fitting. For each sample, we introduced 500 random variations to the average grain size and generated 500 grain-size data. These new grain sizes conform to a log-normal distribution with a standard deviation of 0.02. This distribution is an analogue of the actual grain-size distribution of the sample. Such that, instead of using just one average value for the fitting, a grain-size distribution was used. Examples of the generated grain-size distributions are presented in Figure A10 (a). We conducted 500 fittings based on variation-added grain sizes, resulting in 500 sets of outcomes. The averages of these fittings were then calculated to obtain the final fitting results. Figure A10 (b) show histograms illustrating the results of 500 fits for parameters

$d_0$, $k$, and $n$. The title of each histogram presents the mean value of the corresponding parameter. This method reduces the effect of sample variations in fitting.

Figure A8 present the $n$ values obtained through both methods, while Figure A10 illustrates a comparison between the two fitting curves for pure ice samples at four annealing temperatures. For most samples, the disparity in values of $n$ obtained from the two methods' fitting is within 0.5. However, for samples with a value of $n$ exceeding 6 fitted by the Monte Carlo method, there is a considerable difference in the outcomes of the two methods. The least squares method yields a larger $n$ value, with the difference being at least greater than 1. Moreover, in certain samples where the 3.2-hour grain size data is lower than the initial grain size, the Monte Carlo fitting results range between 1.1 and 5.7, while the least squares fitting outcomes are typically much greater than 20.

However, these samples also exhibit substantial grain growth over longer annealing times, indicating that such large $n$ values are clearly inconsistent with the grain size data. This underscores that, relative to traditional least squares method fitting, the Monte Carlo method can minimize the impact of the sample variations. Compared to the conventional least squares fitting method that relies solely on the average grain size, the Monte Carlo method does not change the original data, but incorporates the grain-size distribution into the fitting process, enhancing the reliability of the fit.

The above discussion is added as an Appendix section.

We have added the following figure and caption to the appendix.

[Figure]

**Appendix Figure A10:** *An example illustrating the Monte Carlo method. Samples of pure ice annealed at 268K are used as examples. 500 sets of grain size data with variation were used for*

*fitting, and corresponding fitting results were obtained. (a) For each duration, 500 sets of variation-added grain size data were generated by applying a range of random noise to the measured average grain size. In each histogram, the red vertical line represents the measured average grain size. (b) Results for parameters $d_0$, k and n are obtained from 500 sets of variation-added grain size. From left to right, three histograms of the parameters, $d_0$, k and n, were illustrated. The title of each histogram presents the mean value of the corresponding parameter.*

**Editor Dr. McCormack**

- Professor Gerbi's raises the following comment about n: "Many places in the manuscript and in the conclusions (e.g., lines 198, 209, 230, 231, 286, 294, 472, 473) n is described as varying." In your response (proposed update to line 232) you indicate that "We found no systematic changes in values of n with changes in solute concentration or temperature." This seems to contradict some earlier statements, e.g. on line 198, n "generally increases with decreasing KCl concentration". Do you put this variation solely to uncertainty in n, rather than any effect of solute concentration? It would be good to add extra commentary on the lines pointed out by Prof Gerbi (e.g., lines 198, 209, 230, 231, 286, 294, 472, 473) around the magnitude of the variation in n, so that it's clear that these are small variations, and to expand the proposed modification of line 232 to justify why the magnitude of variation can be considered to be uncertainty in n.

Reply: Thank you for pointing this out. We apologize for missing the modifications of these lines. After reading Prof. Gerbi's comments, we think that the uncertainties in $n$ could overwrite any trend, if it exists. We would like to remove the statements in lines 191, 198 and 209, and discuss them together in lines 231-233.

Line 191: We decide to remove this sentence "*At 263 K, the value of n increases with decreasing KCl concentration, from 4.6 in samples with $10^{-2}$ mol/L to 5.9 in samples with $10^{-5}$ mol/L.*". The variation truly is insignificant and could mislead readers.

Line 198: Similar here, we remove this sentence "*At 258 K, the value of n also generally increases with decreasing KCl concentration, from 4.9 in samples with $10^{-2}$ mol/L to 6.1 in samples with $10^{-5}$ mol/L.*"

Line 209: Similar here, we remove this sentence "*At 253 K, the value of n also increases with decreasing KCl concentration, from 2.7 in samples with $10^{-2}$ mol/L to 6.0 in samples with $10^{-5}$ mol/L.*"

Line 230: "*At a given impurity concentration, no systematic change of n with temperature is found. At 258 and 253 K, n tends to decrease with increasing impurity concentration in KCl-doped ice; while at 268 and 263 K, no trend of n with concentration is found. In the samples that a trend in n was observed,* **the variations in n are in the same magnitude as the uncertainties in n due to sample-to-sample variation, >1 variation in n (Azuma et al., 2012). While no systematic changes were observed in the value of n as the solute concentration or temperature changed, we cannot rule out that the uncertainty in the value of n may be higher than the systematic changes due to solute concentration and temperature. Thus, at this stage, the average value of n can be regarded as representative of all the samples.*"

Line 286: Prof. Gerbi suggested that the average value of $n$ is not very meaningful. We agree that. Here we remove the average, the sentence becomes "*The values of n determined by least-square fitting for these pure ice samples range from 3.1 to 4.6, as summarized in Table 1*".

Line 294: Same here, the average value of $n$ is not very meaningful. The sentence becomes "*In our study, the values of n are generally greater than both the theoretical value and the values obtained from the above experiments, particularly Azuma et al. (2012)*".

Line 472: Same here, the average of *n* is not very meaningful and does not need to appear in conclusions. "*The values of the best-fit grain growth exponent n lie between 2.6 to 6.2.*"

Line 473: Same here, the average of *n* is not very meaningful and does not need to appear in conclusions. "*The grain growth exponent obtained from pure ice samples ranges from 3.1 to 4.6.*"

- Proposed modification to lines 178, 206 and 223: do you mean "At any given time" or "At a given time"? If the latter, please specify the time.

Reply: Sorry for the confusion. We mean at any given time during the annealing.

Line 178: "*At any given time during the annealing, grain sizes in all KCl-doped ice are coarser than those in pure ice, as illustrated in Figure 6(a).*"

Line 206: "*At any given time, grain sizes in ice doped with higher concentrations ($10^{-2}$ and $10^{-3}$ mol/L) are significantly finer than those in ice doped with lower concentrations ($10^{-4}$ and $10^{-5}$ mol/L), as illustrated in Figure 6(c).*"

Line 223: "*At any given time during the annealing, grain sizes in $MgSO_4$-doped ice are always finer than those in pure ice, as illustrated in Figure 7(b), (c) & (d).*"

- Proposed modification to line 130: does the self-similarity definition apply throughout the time period? Please add this if so.

Reply: Yes. It was used to compare the grain-size distributions at two different time spots during the annealing.

Line 130: "*The normalized standard deviation ($\sigma_d/d$) can be utilized to evaluate the self-similarity of the grain-size distribution throughout the annealing (Faul and Scott, 2006).*"

- Proposed modification to line 232: "relatively independent" is a bit ambiguous. As per earlier comment, it'd be good to specify here that you have considered any variations in n to be due to inherent uncertainties and therefore the average value of n is assumed to be representative for all samples.

Reply: Yes. We have removed the term "relatively independent". Please see the modification above on Line 230.